# A Multiscale Spatiotemporal Model Including a Switch from Aerobic to Anaerobic Metabolism Reproduces Succession in the Early Infant Gut Microbiota

David M. Versluis,[a] Ruud Schoemaker,[b] Ellen Looijesteijn,[b] Daniël Muysken,[c*] Prescilla V. Jeurink,[b] Marcel Paques,[b] Jan M. W. Geurts,[b] Roeland M. H. Merks[a,d]

aLeiden University, Institute of Biology, Leiden, The Netherlands
bFrieslandCampina, Amersfoort, The Netherlands
cCentrum Wiskunde & Informatica, Amsterdam, The Netherlands
dLeiden University, Mathematical Institute, Leiden, The Netherlands

**ABSTRACT** The human intestinal microbiota starts to form immediately after birth and is important for the health of the host. During the first days, facultatively anaerobic bacterial species generally dominate, such as *Enterobacteriaceae*. These are succeeded by strictly anaerobic species, particularly *Bifidobacterium* species. An early transition to *Bifidobacterium* species is associated with health benefits; for example, *Bifidobacterium* species repress growth of pathogenic competitors and modulate the immune response. Succession to *Bifidobacterium* is thought to be due to consumption of intracolonic oxygen present in newborns by facultative anaerobes, including *Enterobacteriaceae*. To study if oxygen depletion suffices for the transition to *Bifidobacterium* species, here we introduced a multiscale mathematical model that considers metabolism, spatial bacterial population dynamics, and cross-feeding. Using publicly available metabolic network data from the AGORA collection, the model simulates *ab initio* the competition of strictly and facultatively anaerobic species in a gut-like environment under the influence of lactose and oxygen. The model predicts that individual differences in intracolonic oxygen in newborn infants can explain the observed individual variation in succession to anaerobic species, in particular *Bifidobacterium* species. *Bifidobacterium* species became dominant in the model by their use of the bifid shunt, which allows *Bifidobacterium* to switch to suboptimal yield metabolism with fast growth at high lactose concentrations, as predicted here using flux balance analysis. The computational model thus allows us to test the internal plausibility of hypotheses for bacterial colonization and succession in the infant colon.

**IMPORTANCE** The composition of the infant microbiota has a great impact on infant health, but its controlling factors are still incompletely understood. The frequently dominant anaerobic *Bifidobacterium* species benefit health, e.g., they can keep harmful competitors under control and modulate the intestinal immune response. Controlling factors could include nutritional composition and intestinal mucus composition, as well as environmental factors, such as antibiotics. We introduce a modeling framework of a metabolically realistic intestinal microbial ecology in which hypothetical scenarios can be tested and compared. We present simulations that suggest that greater levels of intraintestinal oxygenation more strongly delay the dominance of *Bifidobacterium* species, explaining the observed variety of microbial composition and demonstrating the use of the model for hypothesis generation. The framework allowed us to test a variety of controlling factors, including intestinal mixing and transit time. Future versions will also include detailed modeling of oligosaccharide and mucin metabolism.

**KEYWORDS** infant microbiota, microbial ecology, flux balance analysis

Address correspondence to Roeland M. H. Merks, r.m.h.merks@biology.leidenuniv.nl.

*Present address: Daniël Muysken, Vrije Universiteit Amsterdam, Center for Integrative Bioinformatics, Amsterdam, The Netherlands.

The authors declare a conflict of interest. This study was financially supported by FrieslandCampina. R.S., E.L., P.V.J., M.P. and J.M.W.G. are currently or were previously employed by FrieslandCampina.

The human infant microbiota starts forming directly after birth, and it differs greatly from the adult microbiota (1). Three main community types are observed in the infant: (i) a *Proteobacteria*-dominated microbiota; (b) an *Actinobacteria*-dominated microbiota; and (iii) less frequently (12 to 14% in one study [2]), a *Bacilli*-dominated microbiota (1). The three main community types are established after an ecological succession of early communities. This ecological succession is thought to be controlled by nutrition and early oxygen in the colon. To develop hypotheses on the potential mechanisms and controlling factors of the initial development of the human microbiota, in particular the role of early oxygen, we introduced a computational model.

In the first 24 to 48 h after birth, *Proteobacteria*, including *Escherichia coli* and *Enterobacter cloacae*, and *Bacilli*, including *Streptococcus*, *Lactobacillus*, and *Staphylococcus*, are the most common (1, 3). In the following weeks, *Proteobacteria* are often replaced by anaerobic *Actinobacteria*, mainly *Bifidobacterium* species, whereas *Bacilli* are succeeded by either *Proteobacteria* or *Actinobacteria*. Other anaerobic species are also found but typically do not dominate. *Actinobacteria* generally persist as the dominant group until weaning (1). A possible trigger for the replacement of the *Enterobacteriaceae* by *Bifidobacterium* spp. is depletion of a hypothesized initial amount of oxygen (4, 5). Oxygen diffuses into the gut lumen from the body and is taken up by bacteria, by colonocytes, and by nonbiological chemical processes in the cecal contents (6–8), leading to oxygen depletion. The relative importance of these processes is still under debate (7, 8).

*Bifidobacterium* species (1) are associated with a range of health benefits. For example, early succession to a *Bifidobacterium*-dominated microbiota is correlated with reduced probabilities to be underweight at 18 months of age or to experience colic (2, 9). Acetate and lactate produced by *Bifidobacterium* also contribute to the acidification of the infant gut, which suppresses many pathogens (10). A key question, therefore, is if (and how) *Bifidobacterium* can be promoted. The composition and development of the infant microbiota is thought to be determined by many factors, including nutrition (11–13), i.e., human milk or infant formula. The presence of lactose, the most abundant carbohydrate in human milk and most infant formulas, is essential for acquiring high levels of *Bifidobacterium* (14). This suggests that nutrition can play a crucial role in promoting *Bifidobacterium*.

To obtain more insight into the potential ecological and metabolic mechanisms underlying bacterial colonization and succession in the infant colon, here we propose a multiscale, spatial computational model of the infant gut microbiota. Earlier work introduced similar models of the adult gut microbiota (15–18). Those models make use of flux balance analysis (FBA) to model metabolism. Given a number of constraints, including substrate availability and enzyme availability, FBA uses genome-scale metabolic network models (GEMs) to predict optimal biomass or energy production and the exchange of metabolites with the environment. In spatial FBA, FBA models representing small subpopulations of bacteria are coupled together to model a microbial, metabolizing ecosystem. While some approaches assume the whole system is in steady state, allowing use of the optimization approach for the whole ecosystem (16), here we build upon dynamic approaches, such as BacArena (17), COMETS (19), and the approach by Van Hoek and Merks (18). In these models, the metabolite exchange rates given by FBA are used to dynamically change metabolite concentrations in the environment and predict the resulting effects on the bacterial populations. These properties make these dynamic, spatial FBA approaches highly suitable for our purpose of modeling the formation of the infant gut microbiota.

To analyze succession of bacterial species in a dynamic ecosystem such as the infant gut, it is key to accurately describe the dependency of bacterial growth rate and metabolite exchange on environmental metabolite concentrations. Previous work showed that these are well described by constraints on the metabolic capacity of bacterial cells. These are given by an enzymatic constraint which poses a limit on the summed metabolic flux through the whole cell (20), or through more advanced methods, such as flux balance analysis with molecular crowding (FBAwMC) (21), which

weighs the flux constraint according to the efficiency and volume of the individual enzymes (22). Our model is, therefore, based upon the previous approach of Van Hoek and Merks (18), which uses FBAwMC. However, because such detailed data on enzyme efficiency and volume are unavailable for the majority of GEMs, we have replaced FBAwMC with the simpler "enzymatic constraint approach" (20). This approach allows us to model metabolic limitations and substrate-dependent switches in metabolism.

The model represents the first 21 days of development of the infant microbiota in a dynamic, spatially extended, gut-like environment and considers three scales. At the microscopic scale, the model simulates individual metabolites. At the mesoscopic scale, metabolism and growth of bacteria occur as functions of the local availability of metabolites. The local environment is depleted by the bacterial models and receives the metabolites they deposit. This will influence the availability of metabolites in the next timesteps. The macroscopic scale represents the whole ecosystem. At this level, a diffusion model simulates the spread of metabolites between adjacent lattice sites, and a local mixing model simulates mobility of bacterial populations. The metabolites also undergo advection distally to represent luminal movements, while to represent adherence to the mucus the bacterial populations do not undergo advection. Bacterial populations are also randomly deleted to represent local extinction, and they can form new populations in their immediate environment. The system currently considers 15 bacterial species based on 15 GEMs taken from the AGORA collection (23), a database of 773 semiautomatically reconstructed GEMs of human gut bacteria that includes all major infant species (23).

The infant gut microbiota is partially limited by carbohydrates. Prebiotic carbohydrates increase microbial population sizes *in vitro* and in mice inoculated with infant gut bacteria (24, 25). The primary carbohydrate in human milk and nearly all infant formula is lactose, and infant formulas without lactose lead to a different infant microbiota with greatly reduced *Bifidobacterium* abundance (14). Therefore, to a first approximation, we ignored key nutrient sources such as human milk oligosaccharides, fats, protein residues, or intestinal mucus and focused on carbohydrate metabolism, here taking lactose as the only input carbon source in the model.

In comparison with the previous approach (18) upon which the present work builds, the technological advance is that the present framework simulates a dynamic ecology of a large diversity of bacterial species. To this end, the modeling framework can, with minor modifications, import any GEM written in SBML, such as those in the AGORA database that we used in this project (23). The previous system (18) could only describe a "superbacterial" metabolic network representative of the adult microbiota, while bacterial diversity was simulated by activating or repressing individual pathways. In the present work, we studied a selection of species representative of the infant microbiota. The new system also allows us to analyze and visualize the flow of fluxes through the emergent network of dynamically interacting spatially distributed populations. The conceptual advance following from this technological advance is that we can now predict the effects of environmental conditions (oxygen, food, etc.) on the relative abundance of individual bacterial species. This advance has allowed us to use the system to demonstrate the mechanistic consistency of the hypothesis (4, 5) that a transition from an *Enterobacteriaceae*-dominated microbiota to a *Bifidobacterium* spp.-dominated microbiota is explained through gradual depletion of oxygen leading to anaerobiosis. The model suggests that high levels of initial oxygen can lead to prolonged dominance by *Enterobacteriaceae*, even after oxygen has been depleted, possibly explaining observed variations in the composition of the infant microbiota (1, 13), and reveals possible cross-feeding interactions between the species. Altogether, the model proved to be a useful tool for qualitatively evaluating hypotheses on the dynamics of the infant gut microbiota.

## RESULTS

**Model outline.** The model of the infant colon is discussed in detail below in Materials and Methods. Briefly, given a typical infant colon length of 45 cm (26) and diameter of

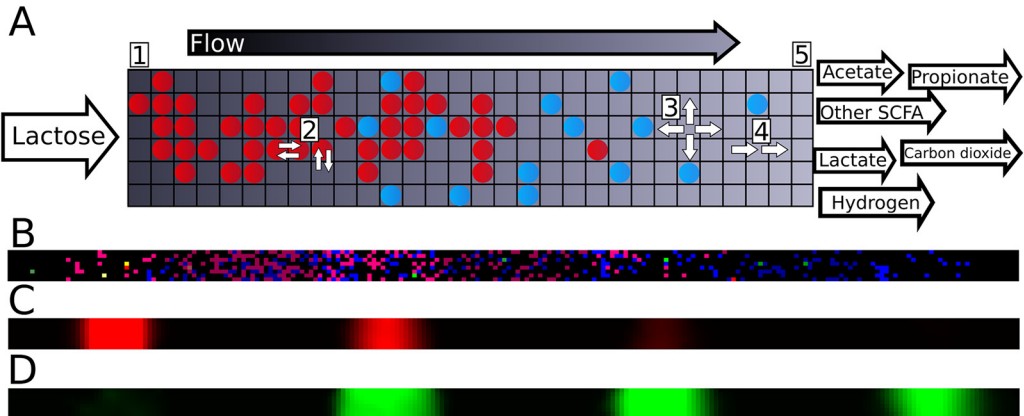

**FIG 1** The multiscale metabolic model. (A) Schematic of the model at a system level. Circles represent bacterial populations, colors represent species. Flow through the tube is from left (proximal) to right (distal). Lactose is placed at the proximal side. Output metabolites are examples and depend on bacterial metabolism. Lattice dimensions and ratios are for schematic purposes. (B) Screenshot of the bacterial layer of the model at a single time point. Colors indicate species, brightness indicates growth rate on the current timestep. (C and D) Screenshots showing lactose (C) and lactate (D) concentrations of the model shown in panel B. Intensity indicates concentration.

approximately 1.5 cm (27), the colon is described on a regular square lattice of 225 × 8 boxes of 2 mm by 2 mm each, thus reaching a good balance between computational efficiency and accuracy (Fig. 1A). Because bacterial populations can vary on a much smaller scale (28, 29), the boxes are purely used as a computational discretization. Each lattice site can contain a population of one species, representing only the locally dominant bacterial species. It is described by a GEM from a list of 15 species prevalent in the earliest infant gut microbiota (11, 30), selected from the AGORA collection (23) (Table 1) (see "Species composition" in the Materials and Methods section).

A simulation proceeded as follows. We initialized the system by placing approximately 540 small populations of the 15 species randomly across the lattice (Fig. 1B). We ran the model in timesteps representing 3 min of real time, and these timesteps proceeded as follows. To mimic carbohydrates entering the colon from the small intestine, a pulse of lactose was introduced every 60 timesteps to the lattice sites of the six most proximal columns (Fig. 1A, step 1). Then, we predicted the metabolism of local bacterial populations by using FBA. To mimic metabolic tradeoffs, we used an enzymatic constraint that set the maximum summed flux for each FBA solution. The uptake bounds for each population were set according to the local concentrations of metabolites (Fig. 1C and D). FBA returns a solution that maximizes the rate of ATP production,

**TABLE 1** Species included in our model[a]

| Species | Family | Anaerobic? (23) | Reactions |
|---|---|---|---|
| *Bacteroides vulgatus* | Bacteroidaceae | Yes | 2474 |
| *Bifidobacterium breve* | Bifidobacteriaceae | Yes | 1987 |
| *Bifidobacterium longum* ssp. *infantis* | Bifidobacteriaceae | Yes | 1005 |
| *Bifidobacterium longum* ssp. *longum* | Bifidobacteriaceae | Yes | 2043 |
| *Blautia hansenii* | Lachnospiraceae | Yes | 2055 |
| *Collinsella aerofaciens* | Coriobacteriaceae | Yes | 920 |
| *Dorea formicigenerans* | Lachnospiraceae | Yes | 2072 |
| *Parabacteroides distasonis* | Porphyromonadaceae | Yes | 2519 |
| *Ruminococcus gnavus* | Lachnospiraceae | Yes | 2225 |
| *Veillonella dispar* | Veillonellaceae | Yes | 1124 |
| *Enterococcus faecalis* | Enterococcaceae | Facultative | 1174 |
| *Enterobacter cloacae* | Enterobacteriaceae | Facultative | 1800 |
| *Escherichia coli* SE11 | Enterobacteriaceae | Facultative | 2356 |
| *Lactobacillus gasseri* | Lactobacillaceae | Facultative | 1163 |
| *Streptococcus salivarius* | Streptococcaceae | Facultative | 1190 |

[a]All species were obtained from the AGORA collection (23). Colors for species correspond to those used in the figures.

**TABLE 2** Extracellular metabolites present in the model in more than micromolar amounts[a]

| Common name | BiGG ID |
| --- | --- |
| Acetate | ac |
| Alpha-ketoglutaric acid | akg |
| Carbon dioxide | co2 |
| Ethanol | etoh |
| Formate | for |
| Hexadecanoate | hdca |
| Hydrogen | h2 |
| L-Lactate | lac_L |
| D-Lactate | lac_D |
| Lactose | lcts |
| Octadecanoate | ocdca |
| Oxygen | o2 |
| Propionate | ppa |
| Proton | h |
| Succinate | succ |
| Tetradecanoate | ttdca |
| Water | h2o |

[a]The BiGG ID codes were obtained from (23).

which is a suitable proxy for the rate of biomass production (31), given the focus on carbohydrate metabolism. This biomass production rate (indicated by brightness in Fig. 1B) was used to update the local population size. Populations spread into an available adjacent lattice site once the population size has exceeded a threshold. The FBA solution also returned a set of metabolite exchange rates, which were used to update the local metabolite concentrations (Fig. 1D). In total, 723 extracellular metabolites were described in the model, but only 17 of these were typically found in more than micromolar amounts in our simulations (Table 2). Bacterial populations diffuse through swaps of the content of adjacent lattice sites (Fig. 1A, step 2). Metabolites diffuse between adjacent lattice sites (Fig. 1A, step 3) and undergo advection to more distal sites (Fig. 1A, step 4) over one lattice per timestep, reaching effective transit times of 11 h (32, 33). Metabolites and populations that reach the distal side of the system (Fig. 1A, step 5) are removed. To mimic the appearance of new bacterial populations (e.g., from ingested spores or introduction from intestinal crypts [34]), each empty lattice site now has a small probability to be filled with a new bacterial population selected at random from all available species (Table 1). This completes the timestep, after which we record the metabolite concentrations and bacterial population in the whole model. All parameters are provided below in Materials and Methods. We also performed a sensitivity analysis for several parameters around the default parameter set shown in Table 3.

We performed several checks to ensure the biochemical and thermodynamic plausibility of the model. After the resulting corrections to the GEMs (see Table S1 in the supplemental material), no atoms were generated or lost by the FBA solutions, there was no flux through the objective function without a substrate, and 99.999% of all fluxes (shown below in Fig. 3A) contained less free energy in the output than in the input. The remaining 0.001% all had no energy increase and growth rates less than 0.001% of the average growth rate. We therefore concluded that the system conserved mass and energy.

**An enzymatic constraint reproduces metabolic switching in *Bifidobacterium*.** Many microorganisms switch between metabolic pathways of higher and lower yield, depending on substrate availability. These so-called metabolic switches can be reproduced with FBA through an enzymatic constraint (20, 22, 35, 36) that limits the summed metabolic flux of a bacterium. *Bifidobacterium* also displays a metabolic switch through a pathway known as the bifid shunt. At low extracellular sugar concentrations, *Bifidobacterium* produces a mixture of acetate, ethanol, and formate in a 1:1:2 ratio from pyruvate. At increased sugar concentrations, the bifid shunt instead reroutes pyruvate to lactate. In addition, acetate is produced from acetyl-phosphate independently, such that at low

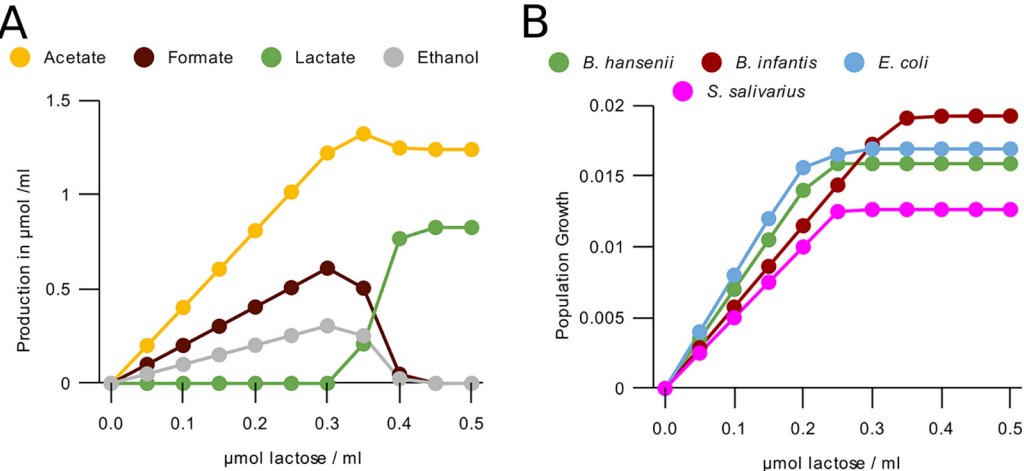

**FIG 2** Enzymatic constraint reproduces metabolic switching in *Bifidobacterium*. (A) Production of metabolites per timestep by a *B. longum* subsp. *infantis* population of 5·10⁹ bacteria with access to one lattice site (0.05 mL) using our FBA with enzymatic constraint method. (B) Growth per timestep by lactose concentration for populations of 5·10⁹ bacteria with access to one lattice site (0.05 mL) of some major bacterial species using our FBA with enzymatic constraint method. *B. longum infantis* is used as the *Bifidobacterium* representative.

concentrations of sugars *Bifidobacterium* produces acetate, ethanol, and formate in a 4:1:2 ratio, and at increased sugar concentrations *Bifidobacterium* switches to production of acetate and lactate in a 3:2 ratio (37, 38).

To test if the enzymatic constraint sufficed to explain the metabolic switch in *Bifidobacterium*, we performed simulations of a single timestep in increasing lactose concentrations (Fig. 2A). These correctly predicted the metabolic switch. The relative production rate of each metabolite (Fig. 2A) before, during, and after the metabolic switch, as well as the pathways used (Fig. S2), matched the experimental observations (37, 38). Simulations without the constraint only reproduced the situation at low concentrations of lactose (see Fig. S1A), showing that the enzymatic constraint is required. The model also predicted an unobserved metabolic switch on lactose in *E. coli* (see Fig. S1B) and reproduced the metabolic switch in *Bacteroides vulgatus* (39) (see Fig. S1C). Another effect of the enzymatic constraint is that the growth rate was saturated at a species-dependent concentration (Fig. 2B; see also Fig. S1D). As a consequence, the relative growth rates of the species depended on the lactose concentration.

**The model predicts *Bifidobacterium* dominance through metabolism under anaerobic conditions.** After having established the effect of the enzymatic constraint in models of individual bacterial populations, we studied the behavior of the full multiscale model for anaerobic conditions. After 21 days (10,080 timesteps), *Bifidobacterium* spp. had become the most abundant in 27 of 30 simulations (Fig. 3A; see also Video S1), a typical situation found in infants of a corresponding age (2, 11). Also in agreement with *in vivo* data, the model predicted the presence of populations of *E. coli*. Our model also predicted populations of *Blautia hansenii*, which is only occasionally present in the infant colon (11). We discuss the niche of this species further in the section "Non-*Bifidobacterium* species benefit from lactate consumption," below. All species other than the aforementioned had abundances either lower or not significantly higher than their initial abundance after 21 simulated days ($P > 0.05$, two-sample *t* test). To test if these species were required for the formation of the community, we ran an additional 30 simulations in which we only included species that had an abundance significantly higher than their initial abundance at the end of other simulations. This led to a very similar set of outcomes (see Fig. S3A). In these simulations, none of the species abundances differed from the original simulations when all species were included ($P > 0.05$, two-sample *t* test). In the remaining simulations, all species were included in order to prevent an initial bias, thus leaving in the possibility that any of the rare species could become abundant under one of the conditions considered. In practice, this did not occur.

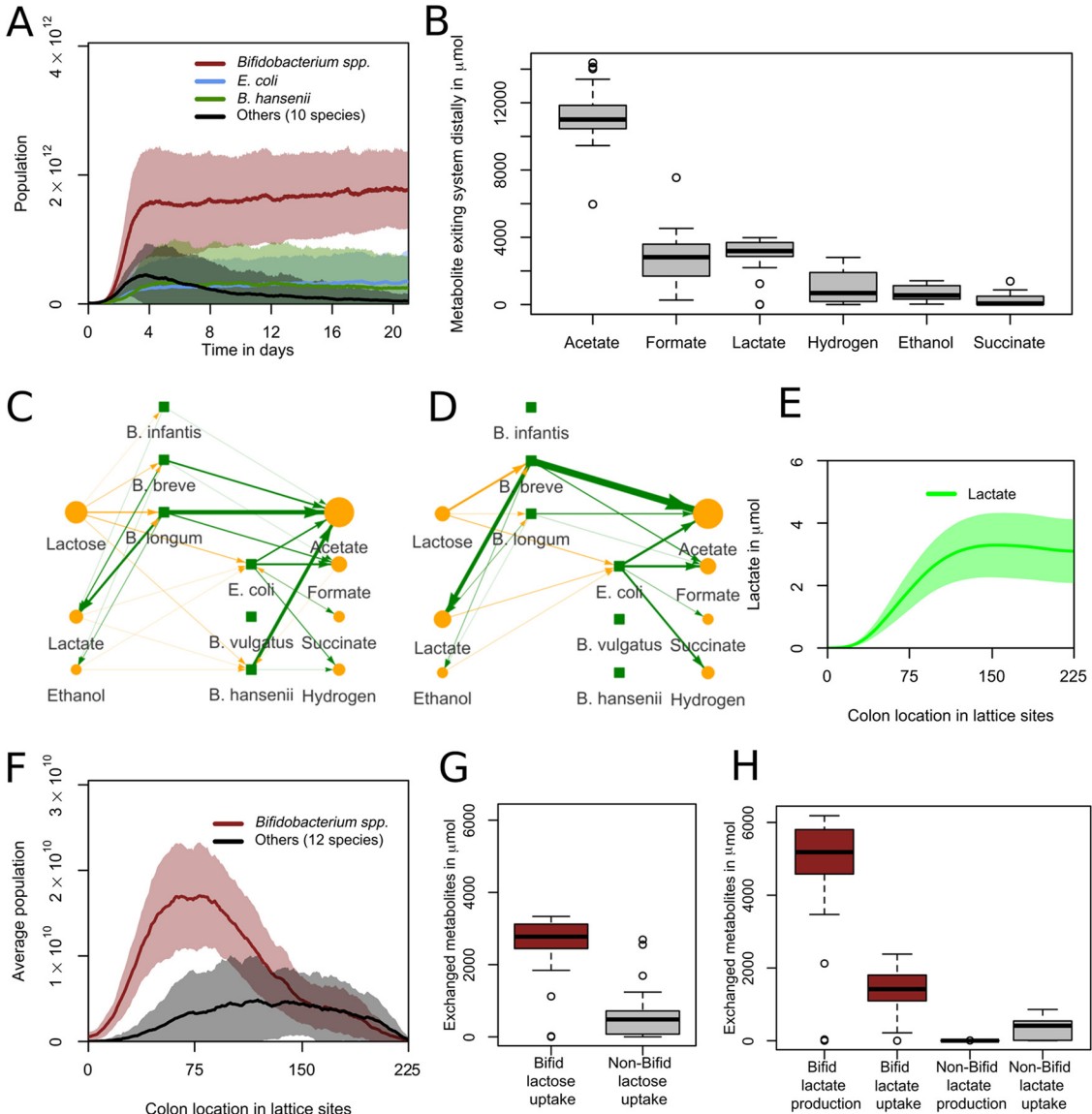

**FIG 3** *Bifidobacterium* spp. become dominant anaerobically and mostly proximally with a unique metabolic profile. (A) Abundances for grouped species over 21 days. One day is 480 timesteps. The curves show mean values, and shaded areas show one standard deviation over $n = 30$ simulations. (B) Distribution of metabolites exiting the system distally over the last 2 days (960 timesteps) of the simulations ($n = 30$). (C and D) Visualization of metabolic interactions in the sample run shown in Video S1. Green lines represent production, and yellow lines represent consumption. Line width and intensity are proportional to the amount exchanged with the environment, with a threshold of 0.5 $\mu$mol, with no normalization. Metabolite circle size is relative to the most abundant metabolite, with a minimum displayed size of 26 pixels. Data are from hour 90 to 93 (step 1,800 to 1,860) (C) and the last 3 h (step 10,020 to 10,080) (D). (E) Spatial distribution of lactate over the last 2 days (960 timesteps) of the simulation. The curves show mean values, and shaded areas show one standard deviation over $n = 30$ simulations. (F) Abundance per location for grouped species over the last 2 days (960 timesteps). The curves show mean values, and shaded areas show one standard deviation over $n = 30$ simulations. (G) Metabolism of lactose by grouped species. All data are from the last 2 days ($n = 30$). (H) Metabolism of lactate by grouped species. All data are from the last 2 days ($n = 30$).

We next analyzed the metabolites diffusing or advecting out of the distal end of the gut over the last 2 days of the simulations (Fig. 3A). We found considerable amounts of acetate, formate, lactate, hydrogen, ethanol, and succinate (Fig. 3B). All other metabolites were present in much smaller quantities. The high abundance of acetate and lower abundance of lactate agreed with the fecal composition of 3-month-old formula-fed infants (40) and was characteristic of microbiota dominated by *Bifidobacterium*. Formate, hydrogen, ethanol, and succinate are also all reported in infant feces (41–44).

To examine how this metabolic pattern is formed, we analyzed the metabolic production and uptake in a sample run (Fig. 3C). *Bifidobacterium* converted lactose into a

mix of acetate, lactate, ethanol, and formate, as predicted (Fig. 2A), with a higher lactate production than ethanol or formate production. Some of this lactate was reabsorbed in later timesteps and converted into the three other metabolites by *Bifidobacterium*. A mixture of lactose and cross-feeding products was converted by *E. coli* and *B. hansenii* into the remaining metabolites shown in Fig. 3B. The main cross-feeding nutrient absorbed was lactate, produced by *Bifidobacterium*, but there was also uptake of ethanol, formate, acetate, and succinate. These were also produced by the cross-feeders. The network became simpler at the end of this run, as *B. hansenii* was at a very low abundance and *E. coli* then only consumed lactose, lactate, and ethanol, which it does not produce (Fig. 3D).

**Cross-feeding emerged in simulations.** The observed cross-feeding in the sample run (Fig. 3C and D) suggested that production of metabolites by *Bifidobacterium* spp. may be crucial for *E. coli* and *B. hansenii*. To further analyze such cross-feeding interactions, we next analyzed the spatial distribution of lactose and the bifidobacterial products lactate and acetate. From day 19 to 21, lactose was present mostly proximally (see Fig. S3B), lactate was most abundant in the middle of the colon (Fig. 3E), and acetate was most abundant distally (see Fig. S3C). To analyze the role of *Bifidobacterium* spp. in shaping this metabolic pattern, we analyzed the locations of populations from day 19 to 21 (Fig. 3F). *Bifidobacterium* spp. were located proximally, whereas the other species were located more distally. These results suggested that separated metabolic niches had formed in the simulated colon. To quantify cross-feeding interactions, we summed the flux of lactose and lactate for *Bifidobacterium* and for all non-*Bifidobacterium* species grouped together. *Bifidobacterium* spp. were the largest consumer of lactose (Fig. 3G) and both the largest producer and consumer of lactate (Fig. 3H). An additional consumption of lactate toward acetate, formate, and ethanol took place at lower lactose concentrations, when the enzymatic constraint was not saturated by lactose uptake. However, the non-*Bifidobacterium* groups consumed more lactate relative to their lactose consumption. To test the effect of lactate uptake by *Bifidobacterium*, additional simulations were run in which the conversion from lactate to pyruvate was blocked, preventing *Bifidobacterium* from consuming lactate. In these simulations, *Bifidobacterium* was the most abundant group in 20 of the 30 simulations, with the largest average abundance (see Fig. S3D). Thus, lactate consumption played a small role but was not the primary cause of *Bifidobacterium*'s dominance. We hypothesized that *Bifidobacterium*'s primary niche is as primary consumer, consuming lactose, while for other species such niches exist as a primary and/or secondary consumer around consuming lactose or lactate. Combined with the observation that lactate concentration was lower in the distal colon (Fig. 3E) and the recorded cross-feeding fluxes observed in our network visualization (see Video S1), we can conclude that the population at the distal end of the simulated colon consists at least partially of such secondary consumers, i.e., cross-feeders.

**Lactose uptake through the bifid shunt is essential for *Bifidobacterium* dominance.** Next, we determined whether the bifid shunt is essential to *Bifidobacterium* dominance in the model. We disabled the bifid shunt by blocking all flux through the reactions associated with fructose-6-phosphate phosphoketolase and ran 30 simulations. Fructose-6-phosphate phosphoketolase is unique to and necessary for the bifid shunt, and it is characteristic of *Bifidobacterium* as a genus (45). *Bifidobacterium* spp. were dominant in none of these simulations (Fig. 4A; see also Video S2). Instead, *E. coli*, *B. vulgatus*, and *B. hansenii* were dominant. This indicated that the bifid shunt is crucial to the dominance of *Bifidobacterium* spp. in our model. We also ran 30 simulations in which only the production of lactate was disabled in *Bifidobacterium* (see Fig. S3F). Twenty-seven of these simulations still led to *Bifidobacterium* dominance. This indicated that the metabolic switch to lactate production was not essential to *Bifidobacterium* dominance in our model. *B. vulgatus* also had a higher average abundance, but this was not significant ($P = 0.27$, two-sample *t* test).

**B. hansenii but not E. coli can sustain itself on cross-feeding.** We observed that non-*Bifidobacterium* species appeared in the simulation behind a proximal population of *Bifidobacterium* spp. (Fig. 3F). These non-*Bifidobacterium* species consumed lactose (Fig. 3D and G), despite the fact that lactose concentrations had dropped compared to the

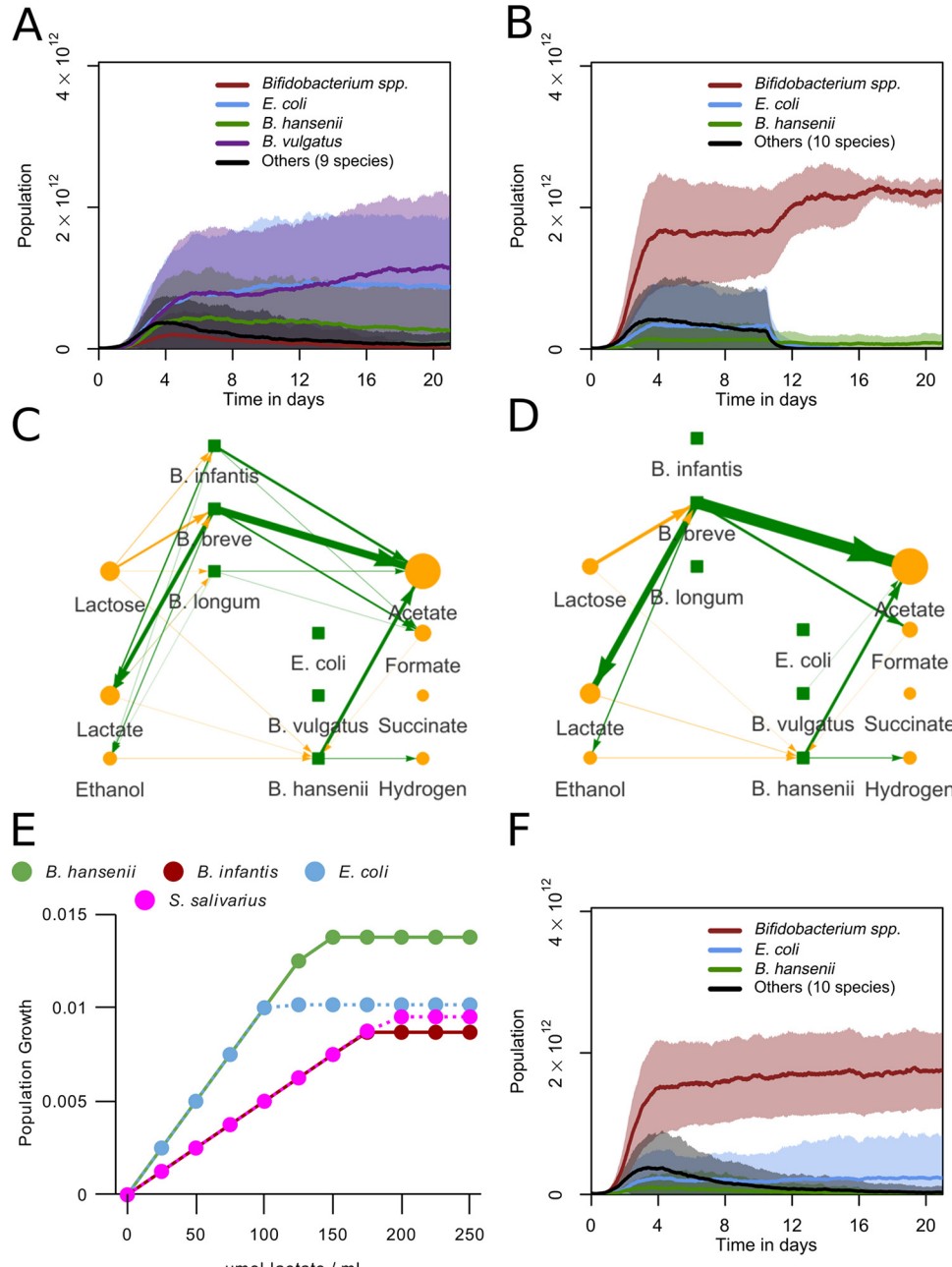

**FIG 4** *Bifidobacterium* and *E. coli* require lactose, but *B. hansenii* requires lactate. (A) Abundances for grouped species over 21 days. The bifid shunt was disabled for all *Bifidobacterium* species. One day is 480 timesteps. The curves show mean values, and shaded areas show one standard deviation over *n* = 30 simulations. (B) Abundances for grouped species over 21 days, but with lactose uptake disabled for non-*Bifidobacterium* species from 5,040 timesteps (10.5 days) onwards. One day is 480 timesteps. The curves show mean values, and shaded areas show one standard deviation over *n* = 30 simulations. (C and D) Visualization of metabolic interactions in a sample run. Green lines represent production, and yellow lines represent consumption. Line width and intensity are proportional to the amount exchanged with the environment, with a threshold of 0.5 $\mu$mol, with no normalization. Metabolite circle size is relative to the most abundant metabolite, with a minimum displayed size of 26 pixels. Data are from hour 90 to 93 (step 1,800 to 1,860) (C) and the last 3 h (step 10,020 to 10,080) (D). (E) Growth per timestep by lactate concentration for populations of 5·10⁹ bacteria with access to one lattice site (0.05 mL) of some major bacterial species. (F) Abundances for grouped species, with lactate consumption disabled for all species. One day is 480 timesteps. The curves show mean values, and shaded areas show one standard deviation over *n* = 30 simulations.

proximal part (see Fig. S3B). We therefore wondered how these non-*Bifidobacterium* species could persist in the model. At reduced lactose concentrations, *E. coli* produced more ATP per timestep than *Bifidobacterium* spp. (Fig. 2B). We hypothesized that non-*Bifidobacterium* species outcompete *Bifidobacterium* at reduced lactose concentrations in the model. To test this hypothesis, we blocked lactose consumption by non-*Bifidobacterium* species after 10.5 days, the midway point of the simulation, and ran 30 simulations. In this way, we ensured that sufficient secondary resources were produced by *Bifidobacterium* in case non-*Bifidobacterium* species made use of them. We observed a sharp population decrease of non-*Bifidobacterium* species starting directly after blocking lactose uptake (Fig. 4B). *E. coli* had a near-zero abundance at the end of these simulations. *B. hansenii* also had a significantly lower abundance compared to the baseline ($P < 0.01$, two-sample $t$ test), but it still had a presence (see Video S2). Thus, in our simulations, some amount of primary consumption was essential for *E. coli*, but not for *B. hansenii*.

**Non-*Bifidobacterium* species benefit from lactate consumption.** To analyze how *B. hansenii* could sustain a population as secondary consumer, we investigated the substrates used by non-*Bifidobacterium* species in the model. Analysis of the flow of metabolites between species in a new run showed that the uptake of lactose by *B. hansenii* decreased over time, while the uptake of lactate increased (Fig. 4C and D). *E. coli* and *B. hansenii* also produced more ATP from lactate than *Bifidobacterium* at any concentration (Fig. 4E). Combined with our earlier observations of cross-feeding on lactate (Fig. 3D and H), we hypothesized that *Bifidobacterium* species cannot compete with *E. coli* and *B. hansenii* on pure lactate. Consumption of lactate by *B. hansenii* could be essential to its ability to sustain populations without lactose uptake.

To investigate if lactate is essential in feeding secondary consumer populations, we blocked the lactate uptake reaction for all species and ran 30 simulations (Fig. 4F; see also Video S2). At 21 days, *B. hansenii* populations were much smaller than in simulations with functional lactate uptake and were not significantly larger than the initial populations ($P > 0.05$, two-sample $t$ test). *E. coli* populations were also significantly smaller than those in simulations with lactate uptake, but they were also significantly larger than their initial population at the start of the run ($P < 0.05$, two-sample $t$ test). *Bifidobacterium* populations were similar to those in the baseline simulations (Fig. 3A). Together with the results in Fig. 4B, these findings indicate that *E. coli* does not require lactate as a cross-feeding substrate as *B. hansenii* does.

**_E. coli_ outcompetes *Bifidobacterium* by taking up early oxygen.** During the first days, *Enterobacteriaceae* or *Bacilli* such as *Streptococcus* are dominant in the *in vivo* infant gut microbiota, after which *Bifidobacterium* spp. are often dominant (1, 11). This pattern may be explained by the presence of oxygen in the gut shortly after birth (3, 5), e.g., via diffusion through the gut wall (7) or accumulation in the newborn infant gut *in utero* in the absence of microbes (46). Intracolonic oxygen diffusion depends on host tissue oxygenation and colonic blood flow (47, 48). To mimic the early presence of oxygen, we introduced 0.1 to 10 $\mu$mol initial oxygen per lattice site. These values were chosen arbitrarily in the absence of precise data, but covered a wide range of outcomes in our model. The oxygen diffused but did not undergo advection in the model.

A 0.1-$\mu$mol level of initial oxygen sufficed to reproduce the pattern observed *in vivo*: *E. coli* initially dominated and *Bifidobacterium* became dominant after a few days or weeks of simulated time (Fig. 5A; see also Video S3). However, we did not see dominance of *Streptococcus* in any of the simulations, in contrast to observations *in vivo* (2). The initial presence of oxygen strongly affected the other species in the model: *B. hansenii* remained nearly entirely absent, while *B. vulgatus* had a higher abundance than it did without initial oxygen. In the network analysis of a sample run, a similar pattern occurred as in the networks shown in Fig. 3C, but with different early species (Fig. 5B). Here, the *Enterobacteriaceae E. coli* and *Enterobacter cloacea* consumed more of the early lactose, along with oxygen. By the end of the run, *Bifidobacterium* spp. again took up the role of primary lactose consumer and lactate producer, whereas *E. coli* fed largely on lactate (Fig. 5C), thus assuming a role of secondary consumer.

At 1 or 10 $\mu$mol initial oxygen per lattice site, *E. coli* and *E. cloacea* remained dominant

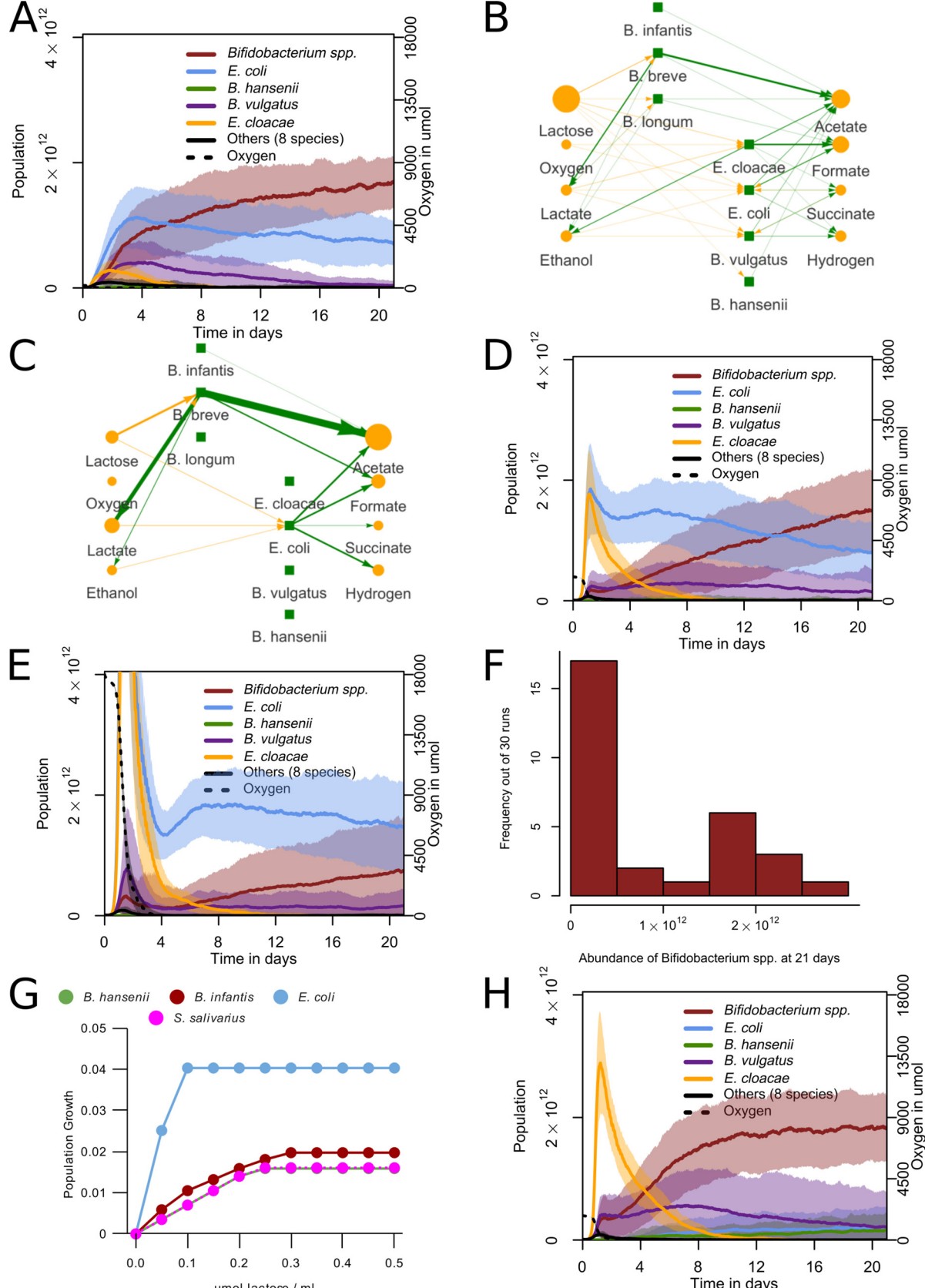

**FIG 5** Initial oxygen leads to initial *E. coli* dominance and succession by *Bifidobacterium*. (A, D, and E) Abundances for grouped species. One day is 480 timesteps. The curves show mean values, and shaded areas show one standard deviation over *n* = 30 simulations. Initial oxygen per lattice

and the main primary consumers for longer (Fig. 5D; see also Video S4), and succession to *Bifidobacterium* took much longer or did not occur at all within the simulated timeframe (Fig. 5E). At 10 $\mu$mol oxygen per lattice site, *Bifidobacterium* either dominated the population or remained much lower in abundance, with few intermediate cases, leading to a bimodal distribution (Fig. 5F). This prediction matched *in vivo* observations (1, 13). The proportion of infants dominated by *Enterobacteriaceae* at 21 days of age varies depending on the study population, but can be 22% (2) or 25% (49) after 21 days; we observed an intermediate value between the 10% (Fig. 5D) and the 60% (Fig. 5E) with different oxygen levels. We also initialized four sets of simulations in which oxygen was released from the upper and lower boundaries of the system with every timestep (see Fig. S4A to D). This led to an increasingly stronger stimulation of *E. coli* with higher levels of oxygen, but it did not lead to any of the temporal effects observed *in vivo*. In a separate set of simulations, we stopped oxygen release at the midpoint of our simulations. We observed an increase of *Bifidobacterium* spp. and a decrease of *E. coli* after that point (see Fig. S4E), leading to a bimodal outcome similar to the simulations with initial oxygen (see Fig. S4F).

We further examined the causes of *E. coli* dominance over *Bifidobacterium* in the presence of oxygen in our model. Fig. 5G shows that when oxygen was available, *E. coli* had much higher growth per timestep in our model than *Bifidobacterium* spp. for all concentrations of lactose, even though *Bifidobacterium* spp. also produced more ATP than observed under anaerobic conditions. *B. hansenii* and *Streptococcus salivarius*, in contrast, grew slower than *Bifidobacterium* for all concentrations in the presence of oxygen. This effect depended on the local oxygen concentration, but *E. coli* grew faster than other species even at concentrations much lower than our initial value (see Fig. S4G). *E. coli* does not use overflow metabolism in our model, though *E. cloacea* does, and *E. coli* does when in the presence of glucose (see Fig. S3H to J).

To test if the direct uptake of oxygen was indeed responsible for *E. coli*'s dominance of the microbiota, we disabled the oxygen uptake reaction of *E. coli* for 30 simulations with 1 $\mu$mol initial oxygen per lattice site. Under these conditions, *E. coli* failed to dominate the population (Fig. 5H). Other species, primarily *E. cloacae*, became dominant early in the simulations as a primary consumer (see Video S4). These species were replaced in all simulations by a population composition similar to that from the simulations without oxygen. In some studies, *Klebsiella* species were the dominant early members of the *Enterobacteriaceae* instead of *E. coli* (50). To examine whether our model could also reproduce early dominance by a *Klebsiella* species, we initialized the simulations with a set of species containing *Klebsiella pneumoniae* instead of *E. coli*. At 0.1 $\mu$mol initial oxygen per lattice site, these simulations behaved the same as the original simulations initialized with *E. coli* (see Fig. S3K): *Bifidobacterium* became the most abundant group after 21 days in 26 out of 30 simulations. In simulations initialized with both *K. pneumoniae* and *E. coli*, *K. pneumoniae* became the most abundant in 15 of 30 simulations, and *Bifidobacterium* became the most abundant in the other 15 (see Fig. S3L). Altogether, the model predicted that *E. coli* relies on direct consumption of oxygen to dominate the microbiota under initially oxygenated conditions in the model. It also indicated that *E. cloacea* does not have an anaerobic metabolism competitive enough to sometimes become dominant over *Bifidobacterium* spp. as *E. coli* does and that other *Enterobacteriaceae* such as *K. pneumoniae* can perform similarly to *E. coli*.

**Sensitivity analysis.** A number of key parameters in our model were based on reasonable estimates. To test the effects of these parameter values on the simulation

**FIG 5** Legend (Continued)

site in was 0.1 $\mu$mol (A), 1 $\mu$mol (D), and 10 $\mu$mol (E). (B and C) Visualization of metabolic interactions in a sample run. Green lines represent production, and yellow lines represent consumption. Line width and intensity are proportional to the amount exchanged with the environment, with a threshold of 0.5 $\mu$mol, with no normalization. Metabolite circle size is relative to the most abundant metabolite, with a minimum displayed size of 26 pixels. Data are from hour 30 to 33 (step 600 to 660) (B) and the last 3 h (step 10,020 to 10,080) (C). (F) Distribution of total *Bifidobacterium* abundance at 21 days (10,080 timesteps) performed with 10 $\mu$mol initial oxygen per lattice site ($n$ = 30 simulations). (G) Growth per timestep by lactose concentration in the presence of abundant oxygen for populations of $5 \cdot 10^9$ bacteria with access to one lattice site (0.05 mL) of some major bacterial species. (H) Abundances for grouped species with oxygen uptake disabled for *E. coli*. The curves show mean values, and shaded areas show one standard deviation over $n$ = 30 simulations.

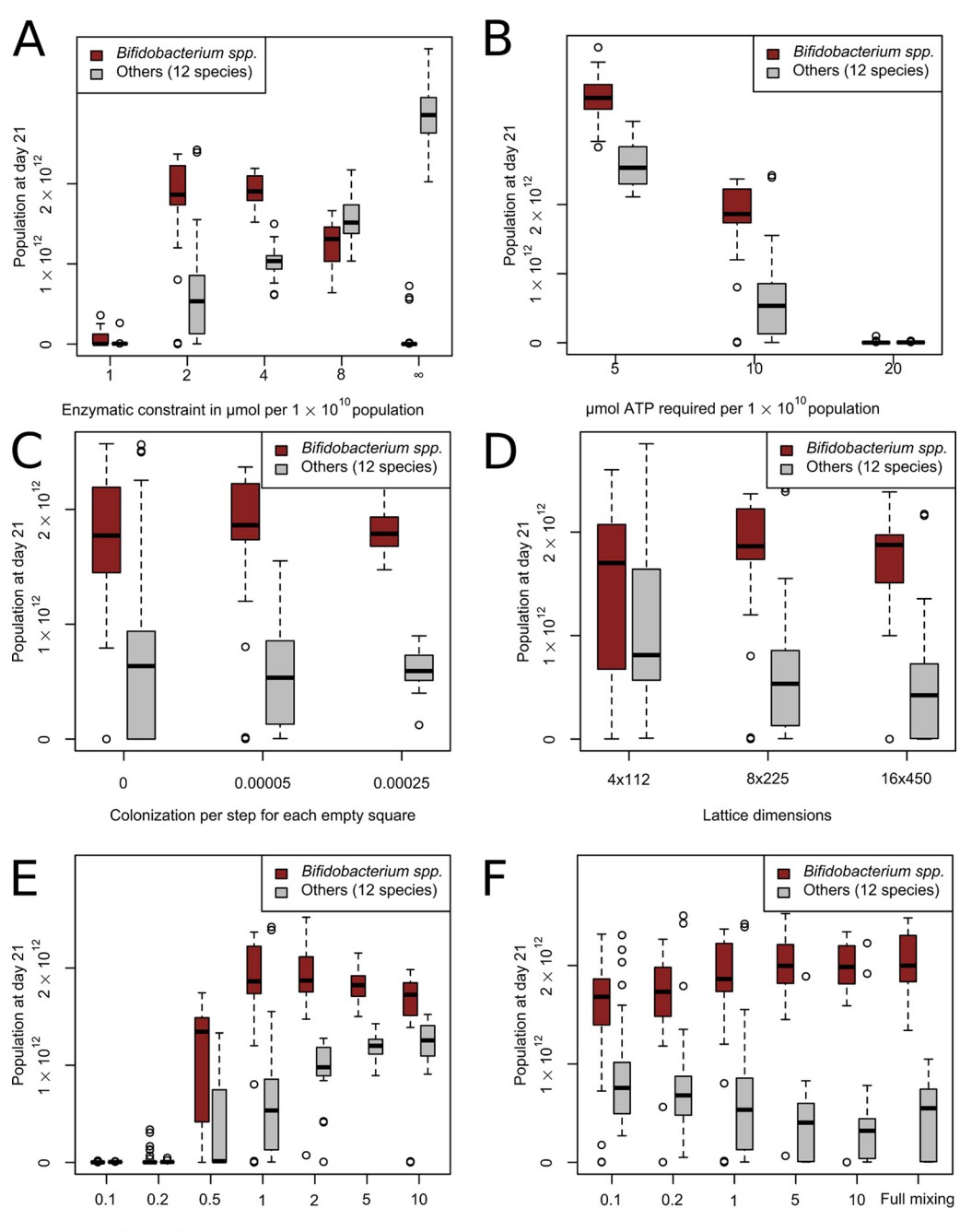

**FIG 6** Sensitivity analysis for estimated parameters. Abundances for *Bifidobacterium* spp. and other species at the end of day 21 with different parameters are shown (*n* = 30 for each condition). (A) Effect of enzymatic constraint around the default value of 2 $\mu$mol flux per timestep per $1 \cdot 10^{10}$ population. (B) Effect of the amount of ATP required for growth of $1 \cdot 10^{10}$ bacteria around the default value of 10 $\mu$mol. (C) Effect of variation in continued influx of populations per timestep per empty square around the default of 0.00005. (D) Effect of variation in width and height of lattice, including adjustment of associated parameters, around the default of 8 × 225 lattice sites. (E) Effect of variation in diffusion for metabolites away from the baseline of $4.4 \cdot 10^{-5}$ cm²/s. (F) Effect of variation in diffusion of bacterial populations away from the baseline of $5.7 \cdot 10^{-5}$ cm²/s.

results, we performed a sensitivity analysis for these parameters under anaerobic conditions, which best represented the situation at the end of the simulations (Fig. 6). As shown in Fig. 2B and Fig. S1D, after relaxing the enzymatic constraint, *Bifidobacterium* spp. lost its competitive advantage at high lactose concentrations (see Video S4). This led to reduced dominance or extinction in whole-gut simulations (Fig. 6A). Tightening the enzymatic constraint led to *Bifidobacterium* spp. remaining dominant, but with a

much smaller population size. Thus, the presence of an enzymatic constraint sufficiently large for metabolic shifts, but not its exact level, is crucial for the prediction of *Bifidobacterium* dominance. We set the enzyme constraint at a level strong enough to predict the metabolic shifts observed *in vitro*, but not so strong as to prevent reasonable growth. The ATP required for a unit of population growth had a linear effect on population size, as expected, but had little effect on the relative abundance of *Bifidobacterium* (Fig. 6B). The large number of populations with a low ATP requirement also made the model computationally unfeasible. The rate of colonization had little effect on the simulation outcomes: at increased colonization rates, *Bifidobacterium* spp. still dominated the microbiota, and its relative abundance remained unchanged (Fig. 6C). Even eliminating colonization altogether hardly affected *Bifidobacterium* spp. dominance in most simulations, but it led to complete extinction of all *Bifidobacterium* species in 5 of 30 simulations and of all 12 non-*Bifidobacterium* species in 9 of 30 simulations. Thus, we kept a moderate amount of colonization in the model. We also examined the effect of placing new populations after initialization only in the first column of the lattice, with a probability increased to match the lower number of eligible squares (see Fig. S5A). This also led to a slightly larger but still comparatively small population for the non-*Bifidobacterium* species. To determine the optimal spatial discretization, we ran the model on a more coarse and on a more refined lattice, scaling the diffusion, advection, and initialization and colonization parameters accordingly (Fig. 6D; see also "Population dynamics" in Materials and Methods). Lattice refinement did not change the dominance of *Bifidobacterium* or the spatial distribution of metabolites and populations (see Fig. S5B to E), but on a more coarse lattice the smaller number of lattice sites seemed to increase the rate of extinctions and variability of metabolite distributions (see Fig. S5F to I). The diffusion of metabolites had a larger effect on the simulation results (Fig. 6E). At low diffusion rates, the populations tended to die out, because they had less opportunity to take up lactose as they were exposed to it only as it "passed by" due to advection. At high diffusion rates, cross-feeders and *Bifidobacterium* consumed all lactate (see Fig. S5J); in infant feces, substantial amounts of lactate are found (40, 51). We therefore decided to choose a value for which the system produced the observed concentration of lactate. Finally, the amount of mixing of bacteria was varied (Fig. 6F). *Bifidobacterium* remained dominant regardless of the diffusion parameter, even when the bacteria were fully mixed (i.e., placed at random locations) for each timestep. Without mixing, the populations could not spread, for lack of empty space. The bacterial diffusion constant was 1.3 times higher than the metabolic diffusion constant. As shown in Fig. 6F, the exact setting of the bacterial diffusion had little effect on the outcomes of the simulations, and so we maintained bacterial diffusion at this baseline.

## DISCUSSION

To develop new hypotheses for possible external effects on the initial phases of the infant gut microbiota, we have developed a dynamic multiscale model. Our simulations reproduced the initial dominance of *Enterobacteriaceae*, particularly *E. coli*, over *Bifidobacterium* species, and the subsequent dominance of *Bifidobacterium* species, out of a broad consortium of species in infants. Moreover, our simulations showed the consistency of the classical hypothesis that this succession could be due to an initial presence of oxygen, followed by microbial depletion (4, 5). These predictions on oxygen suggested that higher levels of initial oxygen may explain the absence or strong delay of *Bifidobacterium* development that is observed in some infants (2). A continuous input of oxygen did not explain the pattern observed *in vivo* (see Fig. S3A to D in the supplemental material), but stopping the input of oxygen at the midway point of the simulation did lead to a similar pattern (see Fig. S3E and F).

While it is known that there is variation in colonic oxygenation (47, 48), the range of this variation in newborns is not known. In premature infants, a positive association exists between the number of days of supplemental oxygen and *Klebsiella* spp.

abundance (50). *Klebsiella* was the primary representative of the *Enterobacteriaceae* in that study population. However, only very few of those infants developed a *Bifidobacterium*-dominated microbiota, regardless of the duration of supplemental oxygen. Nonetheless, it may be desirable to reduce the oxygen concentration faster to achieve the positive health effects of *Bifidobacterium* more consistently. As the direct oxidation of lipids and other organic substrates may contribute to creating anaerobic conditions in the colon (7), altered nutrition may allow for a decrease in oxygen in the colon without stimulating facultative anaerobic bacteria such as *E. coli*. There are also indications that short-chain fatty acids produced by *Bifidobacterium* may decrease oxygen levels by stimulating the oxygen use of colonocytes (8). Future versions of the model could represent these oxygen-depleting mechanisms by localized (e.g., near the intestinal wall) and distributed oxygen consumption terms. At present, the model predicts that distributed oxygen usage by such additional processes will speed up the succession to an anaerobic, *Bifidobacterium*-dominated microbiota. The limit case, i.e., extremely fast oxygen consumption, is represented by the anaerobic case (Fig. 3). The effect of localized oxygen consumption processes, e.g., near the intestinal wall is a topic of our ongoing research.

Our simulations predicted that the dominance of *Bifidobacterium* spp. in the anaerobic infant system could be explained by its consumption of lactose through its bifid shunt metabolism. Our model can correctly represent bifid shunt metabolism due to the implementation of an enzymatic constraint on the maximum total metabolic flux in the bacteria (Fig. 2). Both the low- and high-yield metabolism, and the switch between them based on nutrient availability, have been extensively described *in vitro* (38). The FBA with enzymatic constraint method we use is similar to flux balance analysis with molecular crowding (FBAwMC) (21). As with FBAwMC, we considered the enzymatic constraint to represent the limited availability of space and production capacity for enzymes within a bacterial cell. However, in the absence of good reaction-specific thermodynamic information for our current model, we did not utilize the reaction-specific technique of FBAwMC. We instead based an enzymatic constraint only on the size of the local population, which was sufficient for modeling a metabolic switch. This was equivalent to using FBAwMC and placing all crowding coefficients at the same positive number, which has been done previously to model metabolic switches (21). More broadly, our method is similar to other existing methods that model metabolic switches by including limits on fluxes that represent some physical limit on metabolism, such as proteome constraints or membrane occupancy (35, 52–54). It has been shown that metabolic switches in microbes can be modeled in many ways, provided that two simultaneous constraints are in place (22). In our case, these are the concentration of substrate and the enzymatic constraint.

The metabolic switch of *Bifidobacterium* in our model determined the production of its metabolites. The two most abundant, acetate and lactate, were present in the simulated feces at a median ratio of 3.5:1 (Fig. 3B), similar to that found in formula-fed infant feces (40, 51). We can derive a molar concentration if we assume each of the eight lattice sites whose metabolites are advected out of the system at each step contains 0.05 mL of water. This is based on an estimated total volume of 90 mL, divided over 1,800 lattice sites (Table 3). This resulted in average concentrations of 29 mM acetate and 8 mM lactate in the simulated fecal output analyzed in Fig. 3B. These values were close to, and well within the variability, of the average values for acetate and lactate of 31 and 12 mM, respectively, in the feces of 2-week old infants (51). The median ethanol:acetate ratio of 1:21 is similar to the ratios of 1:18 and 1:10 reported in term infants (43) and the 1:50 reported in preterm infants (55). We used ratios here because the measures per gram dry weight were not available in our model. We observed a lower quantity of ethanol in the simulated fecal output of the simulations where lactate uptake by *Bifidobacterium* was disabled (see Fig. S3E).

*Bifidobacterium* takes up lactate in our model and converts it into pyruvate, catalyzed by lactate dehydrogenase. In our model this pyruvate is then further converted to formate, ethanol, and acetate through the high-yield pathway also used in lactose

**TABLE 3** Default parameters of the model

| Parameter | Value | Units |
|---|---|---|
| Timestep | 180 | s |
| Lattice site side length ($\Delta x$) | 2 | mm |
| Width of lattice | 225 | (No. of lattice sites) |
| Height of lattice | 8 | (No. of lattice sites) |
| Colon transit time | 11 | h |
| Avg no. of initial populations | 540 | |
| Initial size per population | $5 \cdot 10^7$ | (No. of bacteria) |
| Population size required for division | $1 \cdot 10^{10}$ | (No. of bacteria) |
| Death probability | 0.0075 | Per timestep per population |
| Enzymatic constraint | 2 | $\mu$mol flux/timestep/$1 \cdot 10^{10}$ population |
| New species placement probability | 0.00005 | Per timestep per empty lattice site |
| Diffusion of metabolites | $4.4 \cdot 10^{-5}$ | cm$^2$/s |
| Diffusion of populations | $5.7 \cdot 10^{-5}$ | cm$^2$/s |
| Growth per $\mu$mol ATP | $1 \cdot 10^9$ | (No. of bacteria) |
| Lactose input | 211 | $\mu$mol/3 h |

metabolism. However, strong lactate uptake by *Bifidobacterium* spp. has not been observed *in vitro* (56). Thus, while the enzymes used for lactate consumption exist in *Bifidobacterium*, it is not known if the pathway is used *in vivo* to the extent predicted by the model. Lactate uptake by *Bifidobacterium* had only little impact on the simulation results (see Fig. S3D). Similarly, though growth on lactate by *E. coli* in the presence of oxygen is well documented, this is not the case for the anaerobic condition (57). Lactate uptake is not essential for *E. coli*, as it can exist in a more purely primary consumer role (Fig. 4F). The third consumer of lactate, *B. hansenii*, is present only marginally *in vivo* (11). This matches with our results for initially oxygenated conditions, in which *B. hansenii* is also largely absent. However, other species that could fill the same lactate-consuming secondary consumer niche as *B. hansenii* in our model are common *in vivo* (58). Our selection of species may be further improved to allow the correct species to emerge in this secondary consumer niche. For example, it might be appropriate to add *Eubacterium hallii*, a common infant gut bacterium known to consume *Bifidobacterium*-produced lactate and acetate *in vitro* (42).

More generally, our model predicted a lower diversity overall in both bacterial species and metabolites compared to what is observed *in vivo* (11). While the abundant species in our model largely matched with the abundant species *in vivo*, many other species, such as those of the genera *Streptococcus* and *Lactobacillus*, had a lower abundance in our model compared to the *in vivo* conditions (11). We decided to keep all less abundant species in our simulations, to show how in most cases the correct species out of a broad consortium become dominant. There were no large differences when we ran the simulation without these smaller species (see Fig. S3A). The lack of representation of the *Bacilli* species in our model outcomes is particularly notable. While some sources report a portion (e.g., 12 to 14% [2]) of their subjects to be dominated by *Bacilli*, mainly *Streptococcus*, the *Streptococcus* species in our model never became dominant. We showed that *S. salivarius* has an inferior metabolism on lactose and lactate compared to other species in our model (Fig. 2A, 4E, and 5G).

The discrepancy between our model and the *in vivo* data regarding the abundance of less common species such as *Streptococcus* may partially be explained through the focus of our model on carbon dissimilation. Though it would be preferable to represent the whole metabolism of the infant microbiota, including factors such as consumption of amino acids, oligosaccharides, and intestinal mucus, the initial version presented here only considers carbon metabolism from lactose. While a more extensive metabolism might have allowed more mechanisms and niches to be discerned, it would also have introduced additional free parameters, as there are no clear data on the concentration and uptake of these nutrients. The uptake bounds would have to be set arbitrarily, with uniform values or random sampling. Many substances such as fatty acids or protein

residues are not even included as metabolites in the database we use for bacterial metabolism (23), further complicating their introduction to the model. By focusing on carbon metabolism, using ATP production as a proxy for growth rate, and only using lactose as an input nutrient, we can circumvent these problems. ATP production has been shown to be a good proxy for biomass production in *E. coli* (31). Supplementation of the *in vitro* and model organism infant gut microbiota with prebiotic carbohydrates led to a larger microbiota, primarily due to larger *Bifidobacterium* populations (24, 25). This indicates that carbon metabolism is a limiting factor, especially in the absence of prebiotics, as in our model. However, as the increased population in these *in vitro* and *in vivo* studies consisted largely of *Bifidobacterium*, other species were not necessarily also carbon limited. In addition, ATP production may not be a good proxy for biomass production in some species. Species-specific information on the relation between ATP and biomass production may aid future modeling. Finally, gaps in the GEMs we used may have caused ATP production itself to be underestimated compared to what occurs *in vivo*.

Besides details on the modeling of metabolism, the way that diffusion and advection of bacteria and metabolites are handled in the simulations may impact the predictions of our model. The sensitivity analysis demonstrated an unrealistic metabolic output when the metabolic diffusion coefficient was raised to $2.2 \cdot 10^{-4}$ cm²/s (see Fig. S5J). Although this value is much lower than diffusion coefficients measured in the adult intestinal lumen ($1 \cdot 10^{-2}$ cm²/s) (59), in the infant colon mixing may be reduced compared to the adult colon: motor activity that mixes the colon contents (60) is rarer in infants than in older children or adults (61, 62). Exact measurements of mixing in the infant colon will be hard to obtain. Related to this, while we did not examine the effect of bacterial advection in our current work, our previous work (18) suggested that increased bacterial advection greatly reduces diversity and spatial patterning in the model. In fact, compared to the adult gut, in the infant gut there is increased motor activity driving advection (62), and infants display faster colonic transit than adults (32, 33). In light of these data, which seem to indicate much faster bacterial advection and stronger luminal mixing than what we have assumed in our model, a potential interpretation of the present setup of our model is that it represents the dynamics of bacterial populations adhering to the intestinal wall in interaction with metabolites advecting through the lumen. Our future work will explore in more detail how a balance between adhesion of microbiota to the mucus (63) versus advection of bacteria and metabolites in the lumen affects the colonization of the infant colon.

Our modeling approach relates to alternative simulation frameworks, such as Steadycom, BacArena, and COMETS (16, 17, 64, 65). These frameworks, in particular the new COMETS 2 framework (19), would certainly be suitable for answering questions similar to those asked in the present work. In the absence of suitable frameworks at the initial phases of this project, and given the flexibility that comes with using an in-house code base, we preferred to continue developing our own line of gut models (18). A future implementation of our model in one of the available simulation frameworks would be a useful exercise, facilitating future development and comparison of models.

We have used our modeling approach to generate testable hypotheses on the causes and mechanics of succession in the infant gut microbiota, potentially laying the foundation for nutritional interventions that could improve the health of infants. It should be emphasized that the current and future work on this model represent a tool for generating hypotheses and for testing potential mechanical explanations. We cannot fully represent the complexities of the infant gut microbiota, and any generated hypotheses must necessarily be validated *in vitro* and *in vivo*. There is work to be done to further study the relevant factors and to bring these results into a more realistic model context. We aim to do so by integrating the prebiotic oligosaccharide and protein content of nutrition into the model. This may lead to the creation of more niches in the model, and thus a diversity closer to that of the *in vivo* system. We will also continue to improve the selection and curation of metabolic models, such as by disabling

lactate uptake in certain species or including a wider diversity of GEMs. These improvements may lead to novel future insights on the interactions between differences in infant nutrition and succession in the infant microbiota.

## MATERIALS AND METHODS

We used a spatially explicit model to represent the newborn infant microbiota (Fig. 1A). Our model is based on an earlier model of the adult gut microbiota (18). The model consists of a regular square lattice of 225 × 8 lattice sites, where each lattice site represents a space of 2 mm by 2 mm, resulting in an infant colon of 450 mm by 16 mm. Each lattice site can contain any number of metabolites of the 723 types represented in the model, in any concentration, and a single simulated bacterial population. The metabolism of these bacterial populations is based on genome-scale metabolic models (GEMs) from the AGORA collection of species-specific GEMs (23). From this set, we chose 15 GEMs based on a consortium of known infant microbiota species (11, 30). Their metabolic inputs and outputs were calculated using dynamic FBA (66) with an enzymatic constraint functioning as a limit on the total flux through the network (20). The effects of the FBA solution are applied to the spatial environment and then recalculated with each timestep, creating a spatial dynamic FBA.

We identified the two narrow ends of the rectangle with the proximal and distal ends of the colon. In each timestep, metabolites both mix and flow from the proximal to the distal end. Bacterial populations are mixed but do not flow distally as metabolites do.

At the start of each run, we initialized the system with a large number of small populations. We let these perform their metabolism at each timestep. They take metabolites from the environment, and deposit the resulting products. We let the populations grow and divide according to their energy output. Both the initial placement locations and movement of the populations are random, introducing stochasticity in the model. The system develops differently depending on initial conditions, into a diverse and complex ecosystem.

**Species composition.** Each population was represented by a GEM of a species. Fifteen different metabolic models were used in our spatially explicit model (Table 1), and these were selected based on previous research (30). From this list of genera, the most prevalent species within a genus in the vaginally born newborn data set from Bäckhed et al. was used (11). One group from the list could only be determined at the family level: *Lachnospiraceae*. *Ruminococcus gnavus* was chosen to represent this group, based on its high prevalence among this group and the prior inclusion of species from the *Blautia* and *Dorea* genera (67). Because the genus *Bifidobacterium* is known to be particularly diverse (68), we represented it with models of three different strains. All models are based on the GEMs created by Magnúsdóttir et al. in the AGORA collection (23).

**Changes from AGORA.** We use updated versions of the AGORA GEMs (69), to which we applied checks and modifications. First, the objective function was changed from the biomass reaction included in the models to a reaction only requiring ATP production. As this reaction yields only ADP and P$_i$, it was mass neutral. This allowed us to focus on carbon dissimilation within the GEMs and the differences in underlying metabolism. ATP yield has been a good proxy for biomass production in previous studies (31). Focusing on carbon dissimilation meant we could leave all unknown uptake bounds at 0, instead of using an arbitrary or randomized level, as in some other studies (16, 17).

We also checked the metabolic networks for reactions that allow for the occurrence of unrealistic FBA solutions, and we added additional reactions. In the *Bifidobacterium* models, the ATP synthase reaction was made reversible. This allowed it to function as a proton pump, which *Bifidobacterium* species use to maintain their internal pH (70). Lactose permease reactions were added to all *Bifidobacterium* species in the model (71), based on those available in other models in the AGORA collection (23). Combined with the existing reactions in the model and the metabolic constraints, this led to a set of reactions simulating a realistic bifid shunt yield, when substrate was abundant, of 5 mol ATP, 2 mol lactate, and 3 mol acetate per mol of lactose (38, 72). Lactose permease was also added to *Streptococcus salivarius* and *Ruminococcus gnavus* to bring them in line with existing literature on their *in vitro* behavior (73, 74). *Veillonella dispar* was the only species in the model that did not have any lactose uptake (75). A complete list of changes is presented in Table S1 in the supplemental material.

**Checking the validity of the GEMs.** After the changes in Table S1 were applied, all GEMs used in the model were tested individually to ensure that they could grow on a substrate of lactose. Only *Veillonella* did not pass this test, which was consistent with *in vitro* observations (75). *Veillonella* did pass when lactate, a common infant gut metabolite, was used instead. We also tested all GEMs individually for spurious growth in the absence of substrates. To this end, all uptake bounds except water were set to 0. None of the GEMs grew under these conditions. All models were tested for having a net neutral exchange of hydrogen, carbon, oxygen, and nitrogen at each timestep of the model during the simulations. There should be no net change in the number of atoms in the medium due to the calculated fluxes, because the reaction we set as an objective function does not remove any atoms, and there are no other sinks in the simulations. In the present simulation, only water, oxygen, and lactose were introduced into the simulation, so no other atoms than these four were considered. The tests revealed two errors in glycogen metabolism in several GEMs in AGORA that resulted in an energetically favorable removal of intracellular protons from the system. We corrected these GEMs by replacing the reactions responsible for this erroneous energy source (see Table S1). With the corrections we applied to the model, these tests always passed, allowing for rounding errors of less than $1 \cdot 10^{-8}$ $\mu$mol per FBA solution.

The thermodynamic correctness of all reactions was checked by calculating the net difference in Gibbs free energy between input and output metabolites, using a preexisting data set of

thermodynamic source data (76, 77). The conditions assumed for the calculation of the Gibbs free energy were a pH of 7 and an ionic strength of 0.1 M (76). This was recorded for each population at each timestep over the course of a full run of 10,080 timesteps. We determined that 99.999% of all FBA solutions in the simulations of Fig. 3 passed the test by containing less free energy than the associated inputs. The remaining 0.0001% all had an amount of energy in the outputs equal to that of the inputs. All of these solutions had very low growth rates, less than 0.001% of the average growth rate. The sum of these growth rates over 30 simulations was less than 0.05 of an initial population, totaling $2 \cdot 10^6$ bacteria.

**FBA with enzymatic constraint.** For each timestep of the model, a modified version of flux balance analysis with an enzymatic constraint is used by each population to determine what reactions should be used to achieve the most biomass production from the metabolites available to it (20, 78). First, each GEM is converted to a stoichiometric matrix, $S$. All reversible reactions are converted to two irreversible reactions, so that flux is always $\geq 0$. All reactions identified as exchange, sink, or demand in the metabolic reconstruction are marked as exchange in the matrix. These reactions exchange nutrients or metabolites with the environment. For each timestep, all reactions are assumed to be in internal steady state (78):

$$S \cdot \vec{f} = 0, \tag{1}$$

where $\vec{f}$ is a vector of the metabolic fluxes through each reaction in the network (in moles per time unit per population unit).

The vector of fluxes from the environment to the bacterial population (in moles per time unit per population unit) $\vec{F_{in}}$ is constrained by an upper bound, $\vec{F_{ub}}$, which represents the limited availability of most nutrients from the environment:

$$\vec{F_{in}} \leq \vec{F_{ub}}. \tag{2}$$

$\vec{F_{ub}}$ is specified dynamically at each timestep $t$ by the spatial environment (discussed in detail below) at each lattice site $x$:

$$\vec{F_{ub}}(\vec{x}, \vec{t}) = \frac{\vec{c}(\vec{x}, \vec{t})}{B(\vec{x}, \vec{t})}, \tag{3}$$

where $\vec{c}(\vec{x}, \vec{t})$ $\vec{x}$ indicates the location, and $B(\vec{x})$ is the size of the local bacterial population in population units. Population units are continuous, and the size of $B$ can range from $5 \cdot 10^7$ to $1 \cdot 10^{10}$ individual bacteria, or $2 \cdot 10^{10}$ when division is constrained by high density.

There is an additional constraint on the total flux. This constraint represents the limited amount of metabolism that can be performed per cell in each timestep. Each cell can only contain a limited number of enzymes, and each enzyme can only perform a limited number of reactions in a limited time interval. The enzymatic constraint uses the sum of all fluxes $\vec{f}$

$$\sum \vec{f} \leq a. \tag{4}$$

The enzymatic constraint $a$ is in moles per time unit per population unit (Table 3). As both $\vec{f}$ and $a$ are per population unit, this limit scales with population size, allowing each bacterium to contribute equally and independently to the metabolic flux attained in a lattice site. The enzymatic constraint is included as a constraint on each local FBA solution separately at each timestep. The enzymatic constraint was first proposed in a metabolic modeling context in 1990 (20); it was adapted from the study of other capacitated flow networks. In our context, the enzymatic constraint represents the limited availability of physical space for enzymes. Our method is similar to FBA with molecular crowding in that we use an additional constraint to model metabolic capacity (21). However, we did not utilize the necessarily reaction-specific crowding coefficients.

FBA then identifies the solution space that adheres to these constraints, and from this space identifies the solution that optimizes the objective function. The set of solutions consists of a set of exchange fluxes, $\vec{F}(\vec{x}, \vec{t})$, and a growth rate, $g(\vec{x}, \vec{t})$. These exchange fluxes are taken as the derivatives of a set of partial differential equations to model the transport of intermediary metabolites (see below). The size of the population increases in proportion to the growth rate produced by the solution.

**Environmental metabolites.** We modeled a set of 723 extracellular metabolites, the union of all metabolites that can be exchanged with the environment by at least one GEM in the model. In practice, only a fraction of the metabolites occur in any exchange reaction that has flux over it, and only 17 metabolites are ever present in the medium in more than micromolar amounts in our simulations, outside the sensitivity analysis (Table 2). Though the model distinguishes between L-lactate and D-lactate, we display them together in our figures. Nearly all lactate produced and consumed in our model is L-lactate.

To mimic advection, the complete metabolic contents of the lattice except oxygen are moved toward the distal end by 2 mm (one lattice site with the default parameters) per timestep, i.e., once every 3 simulated minutes. This leads to an average transit time of approximately 11 h, in agreement with the observed cecum-to-rectum transit time in newborns (32, 33). Metabolites moving out of the distal end are removed entirely and analyzed separately.

For every timestep, all metabolites diffuse to the four nearest neighbors, $NB(\vec{x})$, at an equal rate for all metabolites (Table 3). We used a baseline of $4.4 \cdot 10^{-5}$ cm$^2$/s to represent mixing of metabolites, which was an order of magnitude higher than normal diffusion for common metabolites (79), to represent active mixing due to colonic contractions. Metabolites were also added and removed by bacterial populations as a result of the FBA solutions, yielding the following equation:

$$\frac{d\vec{c}(\vec{x}, \vec{t})}{dt} = \overrightarrow{F_{out}}(\vec{x}, \vec{t})B(\vec{x}, \vec{t}) - \overrightarrow{F_{in}}(\vec{x}, \vec{t})B(\vec{x}, \vec{t}) + \frac{D}{L^2} \sum_{\vec{i} \in NB(\vec{x})} \left( \vec{c}(\vec{i}, \vec{t}) - \vec{c}(\vec{x}, \vec{t}) \right),$$

(5)

where $F_{out}$ is a vector of fluxes from the bacterial populations to the environment, in moles per time unit per population unit.

All lattice sites initially contain water as their only metabolite, except for the conditions where oxygen is added as well. Metabolites representing the food intake are inserted into the first six columns of lattice sites every 3 h (60 timesteps) to approximate a realistic interval for neonates (80). In the current model, this food intake consists solely of lactose, at a concentration in line with human milk (81), assuming 98% host uptake of carbohydrates before reaching the colon, a commonly used assumption (16). Water is provided as a metabolite in unlimited quantities. Oxygen is placed evenly distributed or at the upper and lower boundaries in some simulations. No other metabolites are available, other than those produced as a result of bacterial metabolism within the model.

**Population dynamics.** Each local population solves the FBA problem based on its own GEM, an enzymatic constraint $a$, its current population size $B(\vec{x}, \vec{t})$, and the local concentrations of metabolites, $\vec{c}(\vec{x}, \vec{t})$, at each timestep, and applies the outcome to the environment (see above) and its own population size, as follows:

$$\frac{dB(\vec{x}, \vec{t})}{dt} = B(\vec{x}, \vec{t})g(\vec{x}, \vec{t}).$$

(6)

Populations at least 200 times the initial size (Table 3) will create a new population in one empty adjacent lattice site, if possible. Half of the old population size is transferred to the new population, in such a way that the total size is preserved. To mimic colonization events, new populations are introduced at random into empty lattice sites during the simulation, representing both dormant bacteria from intestinal crypts (34) and small bacterial populations that are formed from ingested bacteria, which may only become active after having diffused far into the gut. Each empty lattice has a probability of 0.00005 (Table 3) for each step to acquire a new population of a randomly selected species, as follows:

**for** site ∈ latticesites **do**
 **if** site == empty **then**
 **if** randomnumber ≤ placement probability **then**
 Place metabolic network of random species at site.

The probability is scaled by $1/n^2$, with $n$ the scaling factor of $\Delta x$, the side length of a lattice site. There is an equal probability for any species in the model to be selected. As we consider these new populations to be new colonizers, we initialize them at the same population size $B$ as the initial populations in the model (Table 3). Each population dies out at a probability of 0.0075 per timestep, creating a turnover within the range of estimated microbial turnover rates in the mouse microbiota (82).

To mimic the mixing of bacterial populations, the lattice sites swap population contents each timestep. We used the following algorithm, inspired by Kawasaki dynamics (83) and as also used previously for bacterial mixing (18, 84): In random order, the content of each site, i.e., the bacterial population represented by its size and the GEM but not the metabolites, is swapped with a site randomly selected from the set consisting of the site itself and the first- and second-order neighbors. This swap only occurs if both the origin and destination site have not already swapped. Bacterial populations at the most distal column, i.e., at the exit of the colon, are deleted from the system. With this mixing method, the diffusion constant of the bacterial populations becomes $5.7 \cdot 10^{-5}$ cm$^2$/s (Table 3). In the simulations that use a finer or a coarser grid (Fig. 6D), the number of swaps is scaled as $1/n^2$, with $n$ the scaling factor of $\Delta x$, the side length of a lattice site, thus maintaining the same diffusion rate. For an $n$ of <1, all sites are marked as unswapped once all sites have attempted a swap. This allows for sites to swap multiple times. The same approach was used to change the bacterial diffusion rate in the sensitivity analysis (Fig. 6F). To achieve full mixing, all bacteria were assigned random nonoverlapping locations at every timestep. Note that the diffusion rate of bacterial populations with the default parameters (Table 3) was 1.3 times higher than that of the metabolites. As shown by the sensitivity analysis (Fig. 6F), this had little effect on the results.

**Initial conditions.** The simulation was initialized by placing a number of very small populations ($B$) of the various species randomly across different lattice sites of the environment (Table 3). There was a probability of 0.3 for each lattice site to acquire a population, i.e., an average of 540 for our lattice. The size of initial populations was scaled to be roughly equivalent to a plausible initial total load of approximately $3 \cdot 10^{10}$ (85), assuming a total colon volume of approximately 90 mL. As there is little information on the relative abundance of species in the very early infant gut, we placed all species with equal probability. In initially oxygenated conditions, oxygen was also placed as a metabolite. Water was always

considered to be present everywhere. No other metabolites were initially present except for the first feeding.

**Analysis.** At each timestep, we recorded the location and exchange fluxes, $\vec{F}(\vec{x}, \vec{t})$, of major metabolites as well as the size, $B(\vec{x}, \vec{t})$, and species of all populations. This was used to analyze both population composition and metabolic fluxes over time and space. In addition, for each timestep we recorded the location and quantity, $\vec{c}(\vec{x}, \vec{t})$, of all metabolites present in greater than micromolar quantities. We also recorded these metabolite data for all metabolites exiting the system at the distal side. This was considered the closest equivalent to a fecal composition in the model, and these results were compared to data from *in vivo* fecal samples.

To detect any irregularities, we also recorded the net carbon, hydrogen, and oxygen flux of every population and of the system as a whole. The difference in Gibbs free energy per timestep was also recorded per population per FBA solution and separately over the whole system. Estimated Gibbs free energy was derived from the Equillibrator database (71). Energy loss, $l$, in joules per population unit, was recorded as follows, where $i$ is metabolites, $F$ is the exchange fluxes in moles per population unit, and $E$ contains the Gibbs free energy in joules per mole for each metabolite:

$$l = \sum_i F(i) E(i). \tag{7}$$

For specific simulations, reactions were removed from the models. This was performed by deleting some reactions from the GEM before the conversion to the stoichiometric matrix. To remove fructose-6-phosphate phosphoketolase, we removed the reactions R_PKL and R_F6PE4PL. For other simulations, the uptake of certain metabolites was disabled. This was done by placing the upper bound of flux, $\vec{F_{ub}}$, of the relevant exchange reaction at 0 for the relevant populations.

**Parameters.** Relevant parameters are listed in Table 3. Based on measurements of the typical length and diameter of the infant colon (26, 27), we estimated a volume of 90 mL. Combined with the average abundance per milliliter of around $10^{10}$ after the first days (85), this led to a very rough estimate of $10^{12}$ bacteria in the young infant colon. To remain computationally feasible, while still modeling at a high resolution, we only let populations at or above $1 \cdot 10^{10}$ bacteria divide. When dividing, populations split into two equally sized populations. Division only takes place when space is available. Local populations of more than $2 \cdot 10^{10}$, which can only form if no space is available to divide for a longer time period, cease metabolism.

The lactose input is estimated from the known intake of milk, its lactose concentration, and an estimate of precolonic lactose absorption of 98% (16, 86). Little data are available on the growth rate of bacteria within the human colon. Growth rates are expected to be much lower than those found in *in vitro* cultures of individual species (87). In the absence of precise data for infants, here we used a death probability that placed the replacement rate within the range of estimated doubling times of the whole gut microbiota in mice (82). The colonic transit time was based on data for total transit time gathered with carmine red dye (32), adjusted for the mouth-to-cecum transit time (33). The timestep interval was set at 3 min, to be able to capture individual feedings at a high resolution. Values for free parameters were based on estimates and further evaluated in sensitivity analyses.

**Implementation.** The model was implemented in C++11. Our code was based on code developed earlier to model the gut microbiota (18). Random numbers were generated with Knuth's subtractive random number generator algorithm. Diffusion of metabolites was implemented using the forward Euler method. The GEMs were loaded using libSBML 5.18.0 for C++. We used the GNU linear programming kit 4.65 as a linear programming tool to perform each FBA with enzymatic constraint. We used the May 2019 update of AGORA, the latest at the time of writing, from the Virtual Metabolic Human Project website (www.vmh.life). Python 3.6 was used to extract thermodynamic data from the eQuilibrator API (December 2018 update) (71) and to determine mean square displacement of our bacterial diffusion. Model screenshots were made using the libpng16 and pngwriter libraries. Other visualizations and statistical analyses were performed with R 4.1.2 and Google Sheets.

**Data availability.** The code used for the model is available from GitHub at https://github.com/DMvers/IGMOST. The data files used for the model are available from GitHub at https://github.com/DMvers/IGMOSTdatafiles.

## SUPPLEMENTAL MATERIAL

Supplemental material is available online only.

**VIDEO S1**, MOV file, 17.8 MB.
**VIDEO S2**, MOV file, 2.4 MB.
**VIDEO S3**, MOV file, 18 MB.
**VIDEO S4**, MOV file, 3 MB.
**FIG S1**, PDF file, 0.3 MB.
**FIG S2**, PDF file, 1.1 MB.
**FIG S3**, PDF file, 0.4 MB.
**FIG S4**, PDF file, 0.7 MB.
**FIG S5**, PDF file, 0.5 MB.
**TABLE S1**, CSV file, 0.01 MB.

## ACKNOWLEDGMENTS

This study was financially supported by FrieslandCampina. R.S., E.L., P.V.J., M.P., and J.M.W.G. are currently or were previously employed by FrieslandCampina. The work was carried out in part on the Dutch national e-infrastructure with the support of SURF Cooperative. This work was performed in part using the ALICE compute resources provided by Leiden University.

P.V.J., M.P., J.M.W.G., and R.M.H.M. acquired funding. D.M.V., P.V.J., M.P., J.M.W.G., and R.M.H.M. conceived and planned the simulations. D.M.V. and D.M. wrote software used for the simulations. D.M.V. performed the simulations and analyzed the data. R.S., E.L., J.M.W.G., and R.M.H.M. contributed to the interpretation of the results. J.M.W.G. and R.M.H.M. supervised the project. D.M.V. drafted the manuscript. D.M.V., R.S., E.L., J.M.W.G., and R.M.H.M. revised and edited the manuscript.

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
