## [Reviewer comments · mSystems]

A multiscale spatiotemporal model including a switch from aerobic to anaerobic metabolism reproduces succession in the early infant gut microbiota

David Versluis, Ruud Schoemaker, Ellen Looijesteijn, Daniël Muysken, Prescilla Jeurink, Marcel Paques, Jan Geurts, and Roeland Merks

Corresponding Author(s): Roeland Merks, Institute of Biology, Leiden University

Review Timeline:

Submission Date:	May 13, 2022
Editorial Decision:	June 1, 2022
Revision Received:	July 7, 2022
Accepted:	August 2, 2022

Editor: Karoline Faust

Reviewer(s): The reviewers have opted to remain anonymous.

Transaction Report:

DOI: <https://doi.org/10.1128/msystems.00446-22>

June 1, 2022

Prof. Roeland M.H. Merks
Institute of Biology, Leiden University
Animal Sciences & Health
Einsteinweg 55
Leiden 2333 CC
Netherlands

Re: mSystems00446-22 (A multiscale spatiotemporal model including a switch from aerobic to anaerobic metabolism reproduces succession in the early infant gut microbiota)

Dear Prof. Roeland M.H. Merks:

Thank you for resubmitting your manuscript to mSystems. The first reviewer was unable to check your rebuttal, but their comments have in my opinion been adequately addressed. The second reviewer recommends a rejection with resubmission, but given that this work has already undergone a rejection with resubmission and that most of the comments are straightforward to address, I have converted the recommendation into a minor revision. Nevertheless, please thoroughly address the remaining comments of the reviewer and upload the model code to Zenodo as promised.

Below you will find instructions from the mSystems editorial office and comments generated during the review.

Preparing Revision Guidelines

Sincerely,

Karoline Faust

Editor, mSystems

Journals Department
American Society for Microbiology

Reviewer comments:

Reviewer #2 (Comments for the Author):

In the paper 'A multiscale spatiotemporal model including a switch from aerobic to anaerobic metabolism reproduces succession in the early infant gut microbiota', Versluis et al. adapt a digital model of the gut microbiota previously developed in their laboratory to the case of the development of an infant microbiota. The model couples a population dynamic model, including dynamic-FBA metabolic models of prevalent strains of the infant microbiota and a stochastic model of population mixing, to a reaction-diffusion-convection model of metabolites. The paper is well written, the model is correctly described and the numerical experiments make sense. The authors first check that their model of *Bifidobacterium* reproduces a metabolic switch and the use of Bif. shunt at high lactose concentrations, allowing for *Bifidobacterium* dominance under anaerobic conditions and cross-feeding of non-*bifidobacterium* species through lactate. *Bifidobacterium* dominance can be canceled by turning off Bif. shunt. They also show that *B.hansenii* uses lactate while *E.coli* mainly uses lactose in anaerobiosis. Finally, they show that *E.coli* outcompete *Bifidobacterium* under various levels of initial oxygen.

This paper is a re-submission and important updates have been included, compared to the previous version. In particular, the metabolic interactions between species have been investigated much more in depth, and the main part of the methodological issues that were raised have been tackled.

I keep my previous opinion that the main achievement of this work is not technical nor conceptual since the model is the use in another context of a previous work published by the authors. The authors indicate in the response to reviewers that they clarified these achievements at page 4 line 167, but I could not find the referenced clarifications in the text.

Compared to the previous submission, the biological insights are more important, even if, as the author state themselves, they mainly investigate the consistency of the quite well established hypothesis of infant gut microbiota colonization, with primary colonization of facultative anaerobes, followed by obligate anaerobes after oxygen depletion. Despite clear improvements, additional precisions are still needed.

Main remarks:

- 1 - you must explain how the enzymatic constraint parameter has been tuned. This parameter is decisive for the community fate, as indicated by fig 6.A, or suggested by figures 2. A and B.
- 2 - the metabolic switch in *Bifidobacterium* should be explained with more details. In particular, the different pathways activated for the different levels of lactose should be highlighted. I assume that the bif. shunt is activated at all lactose levels, but that an additional consumption of lactate towards acetate, formate and ethanol is allowed at lower lactose concentration, when the enzymatic constraint is not saturated by the sole bif. shunt pathway. If it is the case, it should be explained more clearly.
- 3 - metabolic switches are presented on a lactose gradient. O₂ gradients should also be investigated, since anaerobiosis genesis is presented as the main subject of the paper.
- 4 - why aren't the bacteria advected by the intestinal flow in the model ? (numerous intestinal models have this feature : [10.1073/pnas.1601306113](https://doi.org/10.1073/pnas.1601306113), [10.1016/j.jtbi.2018.12.009](https://doi.org/10.1016/j.jtbi.2018.12.009), [10.1371/journal.pone.0145309](https://doi.org/10.1371/journal.pone.0145309), [10.1186/s12859-020-03923-6](https://doi.org/10.1186/s12859-020-03923-6)).
- 5 - The rationale of random appearance of new bacterial populations should be explained with the same accuracy in the text as in the response to reviewers. This modeling is not natural and it is worth adding explanations.
- 6 - The inclusion of the different mechanisms of O₂ depletion presented in the introduction should be discussed in the discussion.
- 7 - Dynamical networks (in the movie) are usefull. Please clearly state how the normalization of the arrows is done (normalized by the total flux?, by species?). The meaning of the vertices radius is not indicated in the legend. It is however difficult to clearly distinguish quantitative variations in the figures (i.e. fig. 3 c and d, fig. 4 C and D, fig. 5 B and C.): other representations could be intended.
- 8 - distinction between microscopic and mesoscopic scales is not clear to me. Please clarify.
- 9 - line 429 : please further comment why your model could not reproduce *Bifidobacterium* settlement with *Klebsiella* instead of *E.coli*. Do you have an explanation for this feature ?
- 10 - line 466, the authors state that "We use ratios here because the absolute measures, per gram or per milliliter, are not available in our model." However, absolute measures are available : for example fig 2. A indicates micromol per milliliter. Please further comment.
- 11 - explain how the bacteria diffusion is tuned in the sensitivity analysis. Table 3: diffusion of population is not a parameter, but is an indirect feature of your swapping procedure.
- 12 - Do you have an explanation why *B.hansenii* is lower and *E.coli* higher when the 10 "others" species are removed?
- 13 - Could you explain why *B.vulgatus* is as high as *B.hansenii* when lactate production is turned off (fig. S2 F)?

14 - Fig 3. E : y scale has been divided by 10 compared to initial submission. Please comment.

15 - an algorithm should be provided to further supplement the Material and method section "population dynamics".

Minor comments:

- line 263 : wrong reference to fig. 3D. Line 273 : wrong reference to fig. 4B., line 348, wrong reference to fig. S3 Figure A-D.
- Line 400 and 402 : Fig. S4 B-E and Fig. S4 F-I.. line 565, fig 1.B.
- line 296 : Fig. S2 D also shows that turning off lactate consumption in Bifidobacterium leads to increase B.hansenii, which is an additional support for lactate consumption.
- Figure legend were not provided for supplementary figures. It was difficult to review.
- Please explain why you mentioned overflow line 360.
- typos : line 392, 631.
- streptococcus results should be introduced before line 500, during fig. 2.B or fig. 4.E descriptions, where they are introduced.
- indicate the time integration scheme (why do you have instability for diffusion?)
- eq. 5 : as F is always positive in your formalism, this equation is for produced metabolite only. A minus sign should be added for consumed metabolites.

In the paper 'A multiscale spatiotemporal model including a switch from aerobic to anaerobic metabolism reproduces succession in the early infant gut microbiota', Versluis et al. adapt a digital model of the gut microbiota previously developed in their laboratory to the case of the development of an infant microbiota. The model couples a population dynamic model, including dynamic-FBA metabolic models of prevalent strains of the infant microbiota and a stochastic model of population mixing, to a reaction-diffusion-convection model of metabolites. The paper is well written, the model is correctly described and the numerical experiments make sense. The authors first check that their model of Bifidobacterium reproduces a metabolic switch and the use of Bif. shunt at high lactose concentrations, allowing for Bifidobacterium dominance under anaerobic conditions and cross-feeding of non-bifidobacterium species through lactate. Bifidobacterium dominance can be canceled by turning off Bif. shunt. They also show that B.hansenii uses lactate while E.coli mainly uses lactose in anaerobiosis. Finally, they show that E.coli outcompete Bifidobacterium under various levels of initial oxygen.

This paper is a re-submission and important updates have been included, compared to the previous version. In particular, the metabolic interactions between species have been investigated much more in depth, and the main part of the methodological issues that were raised have been tackled.

I keep my previous opinion that the main achievement of this work is not technical nor conceptual since the model is the use in another context of a previous work published by the authors. The authors indicate in the response to reviewers that they clarified these achievements at page 4 line 167, but I could not find the referenced clarifications in the text.

Compared to the previous submission, the biological insights are more important, even if, as the author state themselves, they mainly investigate the consistency of the quite well established hypothesis of infant gut microbiota colonization, with primary colonization of facultative anaerobes, followed by obligate anaerobes after oxygen depletion. Despite clear improvements, additional precisions are still needed.

Main remarks:

- 1 - you must explain how the enzymatic constraint parameter has been tuned. This parameter is decisive for the community fate, as indicated by fig 6.A, or suggested by figures 2. A and B.
- 2 - the metabolic switch in Bifidobacterium should be explained with more details. In particular, the different pathways activated for the different levels of lactose should be highlighted. I assume that the bif. shunt is activated at all lactose levels, but that an additional consumption of lactate towards acetate, formate and ethanol is allowed at lower lactose concentration, when the enzymatic constraint is not saturated by the sole bif. shunt pathway. If it is the case, it should be explained more clearly.

3 - metabolic switches are presented on a lactose gradient. O₂ gradients should also be investigated, since anaerobiosis genesis is presented as the main subject of the paper.

4 - why aren't the bacteria advected by the intestinal flow in the model ? (numerous intestinal models have this feature :

10.1073/pnas.1601306113, 10.1016/j.jtbi.2018.12.009,
10.1371/journal.pone.0145309, 10.1186/s12859-020-03923-6).

5 - The rationale of random appearance of new bacterial populations should be explained with the same accuracy in the text as in the response to reviewers. This modeling is not natural and it is worth adding explanations.

6 - The inclusion of the different mechanisms of O₂ depletion presented in the introduction should be discussed in the discussion.

7 - Dynamical networks (in the movie) are usefull. Please clearly state how the normalization of the arrows is done (normalized by the total flux?, by species?). The meaning of the vertices radius is not indicated in the legend. It is however difficult to clearly distinguish quantitative variations in the figures (i.e. fig. 3 c and d, fig. 4 C and D, fig. 5 B and C.): other representations could be intended.

8 - distinction between microscopic and mesoscopic scales is not clear to me. Please clarify.

9 - line 429 : please further comment why your model could not reproduce Bifidobacterium settlement with Klebsiella instead of E.coli. Do you have an explanation for this feature ?

10 - line 466, the authors state that "We use ratios here because the absolute measures, per gram or per milliliter, are not available in our model." However, absolute measures are available : for example fig 2. A indicates micromol per milliliter. Please further comment.

11 - explain how the bacteria diffusion is tuned in the sensitivity analysis. Table 3: diffusion of population is not a parameter, but is an indirect feature of your swapping procedure.

12 - Do you have an explanation why B.hansenii is lower and E.coli higher when the 10 "others" species are removed?

13 - Could you explain why B.vulgatus is as high as B.hansenii when lactate production is turned off (fig. S2 F)?

14 - Fig 3. E : y scale has been divided by 10 compared to initial submission. Please comment.

15 - an algorithm should be provided to further supplement the Material and method section "population dynamics".

Minor comments:

- line 263 : wrong reference to fig. 3D. Line 273 : wrong reference to fig. 4B., line 348, wrong reference to fig. S3 Figure A-D. Line 400 and 402 : Fig. S4 B-E and Fig. S4 F-I.. line 565, fig 1.B.

- line 296 : Fig. S2 D also shows that turning off lactate consumption in Bifidobacterium leads to increase B.hansenii, which is an additional support for lactate consumption.

- Figure legend were not provided for supplementary figures. It was difficult to review.

- Please explain why you mentioned overflow line 360.

- typos : line 392, 631.

- streptococcus results should be introduced before line 500, during fig. 2.B or fig. 4.E descriptions, where they are introduced.

- indicate the time integration scheme (why do you have instability for

diffusion?)

- eq. 5 : as F is always positive in your formalism, this equation is for produced metabolite only. A minus sign should be added for consumed metabolites.

Response to reviewers (line numbers to marked-up manuscript)

David Versluis and Roeland Merks

July 2022

Response to reviewer 2

In the paper 'A multiscale spatiotemporal model including a switch from aerobic to anaerobic metabolism reproduces succession in the early infant gut microbiota', Versluis et al. adapt a digital model of the gut microbiota previously developed in their laboratory to the case of the development of an infant microbiota. The model couples a population dynamic model, including dynamic-FBA metabolic models of prevalent strains of the infant microbiota and a stochastic model of population mixing, to a reaction-diffusion-convection model of metabolites. The paper is well written, the model is correctly described and the numerical experiments make sense. The authors first check that their model of Bifidobacterium reproduces a metabolic switch and the use of Bif. shunt at high lactose concentrations, allowing for Bifidobacterium dominance under anaerobic conditions and cross-feeding of non-bifidobacterium species through lactate. Bifidobacterium dominance can be canceled by turning off Bif. shunt. They also show that *B.hansenii* uses lactate while *E.coli* mainly uses lactose in anaerobiosis. Finally, they show that *E.coli* outcompete Bifidobacterium under various levels of initial oxygen.

This paper is a re-submission and important updates have been included, compared to the previous version. In particular, the metabolic interactions between species have been investigated much more in depth, and the main part of the methodological issues that were raised have been tackled.

Thank you for your comments

I keep my previous opinion that the main achievement of this work is not technical nor conceptual since the model is the use in another context of a previous work published by the authors. The authors indicate in the response to reviewers that they clarified these achievements at page 4 line 167, but I could not find the referenced clarifications in the text.

*It turned out that, as the reviewer correctly points out, the advances discussed in our first rebuttal letter were not yet fully integrated into the revised manuscript. We apologize for this oversight and have corrected this in the present revision at **page 4, line 140**.*

Briefly, in the present work we have extended the simulation framework developed in Ref. [2] to allow for the import of individual bacterial species (replacing the previous evolving 'superbacterium' approach). This has allowed us to move from a relatively abstract 'toy' model to a more specific model considering actual species, not only metabolic pathways. We apply the model to provide mechanistic, biological insight into the development of the infant gut microbiota. To us it seems not unreasonable nor uncommon that a research team builds upon their previous technical contributions in order to develop new conceptual advancements and achieve new biological insight.

The technical and conceptual achievements compared to the previous work are, in short:

- Modelling an ecological network of individual, actual bacterial species taken from the AGORA database, with metabolic constraints that allow for metabolic switches to exist, as opposed to the previous, more abstract, 'superbacterium' approach*
- Analyzing and visualising the flow of fluxes through a network of dynamically interacting spatially distributed populations*

*These are described at **page 4, line 140***

Compared to the previous submission, the biological insights are more important, even if, as the author state themselves, they mainly investigate the consistency of the quite well established hypothesis of infant gut microbiota colonization, with primary colonization of facultative anaerobes, followed by obligate anaerobes after oxygen depletion. Despite clear improvements, additional precisions are still needed.

Main remarks:

1 - you must explain how the enzymatic constraint parameter has been tuned. This parameter is decisive for the community fate, as indicated by fig 6.A, or suggested by figures 2. A and B.

We have selected the enzymatic constraint parameter to be within the range of settings where the key species reproduce the metabolic shifts observed in vitro (Fig. 2 and Fig. S1). Even if quantitative values of this constraint are unknown, it is possible to make reasonable estimates based on the model predictions, followed by a detailed evaluation of the consequences of that particular parameter selection, which is the strategy that we have followed. Put simply, if the constraint is too weak, the model always predicts optimization of yield as in regular FBA. For a range of more stringent conditions metabolic shifts are predicted, and the model then consistently predicts dominance of Bifidobacteria as shown in the sensitivity analysis (Figure 6A). For the simulations outside of the sensitivity analysis we have selected a medium value of the enzymatic constraint,

strong enough to predict the metabolic shifts observed in vitro, but not so strong as to prevent reasonable growth.

*We have clarified this at **page 10, line 452.***

2 - the metabolic switch in Bifidobacterium should be explained with more details. In particular, the different pathways activated for the different levels of lactose should be highlighted. I assume that the bif. shunt is activated at all lactose levels, but that an additional consumption of lactate towards acetate, formate and ethanol is allowed at lower lactose concentration, when the enzymatic constraint is not saturated by the sole bif. shunt pathway. If it is the case, it should be explained more clearly.

*The bifid shunt is indeed active for all lactose levels. The metabolic switch that occurs between high concentration and low concentration lactose, as depicted in Fig. 2, hinges around pyruvate being converted into either lactate or Acetyl-CoA, as the natural system does. This switch is also depicted in, for example, Fig. 1 of [1]. We explain this pathway at **page 5, line 216** and now discuss how the pathways used before and after the metabolic switch correspond to the in vivo situation at **page 6, line 228**. We list the reactions used in a low-concentration, medium-concentration and high-concentration of lactose in the new S2 Table.*

*The reviewer is correct in suggesting that in the full model Bifidobacterium converts lactate into formate, ethanol, and acetate if lactose is scarce. This only occurs when the enzymatic constraint is not saturated by lactose uptake and there is lactate present extracellularly. This effect is not observed in in vitro experiments as discussed at **page 13, line 569**. We have clarified lactate consumption at **page 7, line 291**.*

3 - metabolic switches are presented on a lactose gradient. O₂ gradients should also be investigated, since anaerobiosis genesis is presented as the main subject of the paper.

*We have included these for major species in supplemental figure 3G and discuss them briefly at **page 10, line 420**. In short, E. coli benefits much more from the increased oxygen concentration than Bifidobacterium or other species.*

4 - why aren't the bacteria advected by the intestinal flow in the model ? (numerous intestinal models have this feature : 10.1073/pnas.1601306113, 10.1016/j.jtbi.2018.12.009, 10.1371/journal.pone.0145309, 10.1186/s12859-020-03923-6).

At present the bacterial populations in the model do not advect, and we do agree with the reviewer that this may affect the results. We have studied the effect of having fast advection in previous work [2]. Here advection reduced the diversity and spatial separation in the system. In light of data suggesting faster bacterial advection and stronger luminal mixing than what we have assumed in

*our model, the present set-up of our model is interpreted as the dynamics of bacterial populations adhering to the intestinal wall in interaction with metabolites advecting through the lumen. We discuss this at **page 14, line 629**.*

5 - The rationale of random appearance of new bacterial populations should be explained with the same accuracy in the text as in the response to reviewers. This modeling is not natural and it is worth adding explanations.

*We have added further clarification at **page 20, line 888**.*

6 - The inclusion of the different mechanisms of O₂ depletion presented in the introduction should be discussed in the discussion.

*We have added additional discussion of these mechanisms in the context of our results to the discussion at **page 12, line 521**. In short, future versions of the model could represent these oxygen-depleting mechanisms through additional oxygen consumption terms. The model predicts that distributed oxygen usage by such additional processes will speed up the succession. In the limit case of much faster depletion due to colonocytes or non-biological factors, the situation would be identical to our anaerobic models, where we show that no succession occurs and *Bifidobacterium* is the first dominant group.*

7 - Dynamical networks (in the movie) are usefull. Please clearly state how the normalization of the arrows is done (normalized by the total flux?, by species?). The meaning of the vertices radius is not indicated in the legend. It is however difficult to clearly distinguish quantitative variations in the figures (i.e. fig. 3 c and d, fig. 4 C and D, fig. 5 B and C.): other representations could be intended.

We have now clarified these points in the figure captions. In short: the width and intensity of the lines is directly proportional to the total amount of metabolite in moles exchanged with the medium over the 60 timestep timeframe of each image, and not normalized. The threshold for display is 0.5 μmol of flux over the timeframe. The radius of the circles is normalised to the largest metabolite pool. This pool is 75 pixels. Each other pool is normalised as a portion of this pool, with a minimum size of 26 pixels to maintain visibility. The visual size of bacterial populations is fixed to maintain visibility.

8 - distinction between microscopic and mesoscopic scales is not clear to me. Please clarify.

*We have clarified the description of the scales at **page 3, line 112**. In short, we consider the microscopic scale to represent individual metabolites and reactions, while the mesoscopic scale operates at the scale of individual bacterial populations within single lattice sites.*

9 - line 429 : please further comment why your model could not reproduce Bifidobacterium settlement with Klebsiella instead of E.coli. Do you have an explanation for this feature ?

*This is simply because Klebsiella was not included in the list of species, because it was not prevalent in the experimental study that we have used to select the list of species to be used in the model. However, the reviewer correctly alludes to the fact that Klebsiella is, in fact, sometimes present at high quantities in the infant microbiota [5], which is a good reason to include it in the model simulations. To determine if the model predicts the same succession pattern with Klebsiella we ran an additional set of 30 simulations with Klebsiella pneumoniae instead of E. coli. The succession pattern largely parallels that observed for E. coli (Fig. S3K): K. pneumoniae is initially dominant, and maintains a high abundance, but Bifidobacterium is more abundant in 27 out of 30 runs after 21 simulated days. We also ran a set of 30 simulations where both K. pneumoniae and E. coli were present. These led to dominance of K. pneumoniae over E. coli. Bifidobacterium had the highest abundance in 15 out of these 30 runs. We describe these results at **page 10, line 431**, and thank the reviewer for this helpful suggestion.*

10 - line 466, the authors state that "We use ratios here because the absolute measures, per gram or per milliliter, are not available in our model." However, absolute measures are available : for example fig 2. A indicates micromol per milliliter. Please further comment.

*Thank you for pointing this out. If we assume that the 8 lattice sites that advect out of the gut each timestep contain 0.05 ml of water each, as assumed in Fig. 2A, we can indeed obtain absolute concentrations in the simulated fecal output. This rough estimate of 0.05 ml per lattice site is based on the estimated total volume of the colon (90ml), divided by the number of lattice sites (1800). For acetate and lactate the absolute measure is close to that of the cited paper that provided absolute measures per ml [4]. Other metabolites were rare or not measured in this dataset [4]. The other cited papers provided measurements per gram dry weight [7, 6] or in arbitrary units [3], precluding direct comparison with the simulation predictions. We have described the comparison at **page 12, line 555**.*

11 - explain how the bacteria diffusion is tuned in the sensitivity analysis. Table 3: diffusion of population is not a parameter, but is an indirect feature of your swapping procedure.

*We have further explained how the bacterial diffusion rates were selected in the sensitivity analysis in the methods at **page 20, line 930**. The bacterial diffusion rates indeed follow from the swapping procedure and are measured through the mean squared displacement of particles. To reduce the diffusion rate, the number of swaps per timestep is reduced, while the swapping procedure is it-*

erated a few times for increased values of the bacterial diffusion rate. Complete mixing assigns each population a random location. The paragraph on metabolite diffusion and bacterial diffusion (**page 20, line 907**) has also been rewritten for clarity.

12 - Do you have an explanation why *B.hansenii* is lower and *E.coli* higher when the 10 "others" species are removed?

*The differences between the two simulation conditions are not statistically significant ($p=0.83$ and $p=0.35$, respectively, two-sided t-tests). We now discuss this at **page 6, line 251**.*

13 - Could you explain why *B.vulgatus* is as high as *B.hansenii* when lactate production is turned off (fig. S2 F)?

*After further data analysis we found that the increased *B. vulgatus* abundance is not significant compared to the baseline ($p=0.27$, two-sided t-test). We have noted this at **page 8, line 321**.*

14 - Fig 3. E : y scale has been divided by 10 compared to initial submission. Please comment.

The original y-axis label on Fig. 3E was incorrect, and had been corrected. We have also corrected the x-axis label on Fig 4E in our new version. The previous values were per 100ml, instead of per ml as stated. Fig 6F has been amended to include the correct data for the 5x diffusion condition. Previously, the data for the x10 condition was printed twice. None of these corrections have impacted the results or conclusions.

15 - an algorithm should be provided to further supplement the Material and method section "population dynamics".

*We have now added an explicit algorithm to describe the colonization at **page 20, line 893**.*

Minor comments: - line 263 : wrong reference to fig. 3D. Line 273 : wrong reference to fig. 4B., line 348, wrong reference to fig. S3 Figure A-D. Line 400 and 402 : Fig. S4 B-E and Fig. S4 F-I.. line 565, fig 1.B.

Thank you, all have been corrected

- line 296 : Fig. S2 D also shows that turning off lactate consumption in *Bifidobacterium* leads to increase *B.hansenii*, which is an additional support for lactate consumption.

Thank you for pointing this out. The apparant increase of B.hansenii, is not significant (p=0.33, two-sided t-test).

- Figure legend were not provided for supplementary figures. It was difficult to review.

We apologize for the inconvenience. Unfortunately, mSystems requires the supplementary figure legends at the bottom of the manuscript text, and not included in the supplementary figure pdfs. The supplementary figure legends are located between the methods and references.

- Please explain why you mentioned overflow line 360.

This addressed a comment from the other reviewer asking whether metabolic switches occurred for species other than Bifidobacterium and under other conditions.

- typos : line 392, 631.

Thank you, typos were corrected

- streptococcus results should be introduced before line 500, during fig. 2.B or fig. 4.E descriptions, where they are introduced.

*We have amended the text and now mention the in vivo presence of Streptococcus in the results at **page 9, line 369** and the lack of Streptococcus in our model results even with oxygen at **page 9, line 386**.*

- indicate the time integration scheme (why do you have instability for diffusion?)

*We use the forward Euler method for our metabolic diffusion, which we now mention at **page 22, line 1016**. The paragraph on metabolite diffusion and bacterial diffusion (**page 20, line 907**) has also been rewritten for clarity.*

- eq. 5 : as F is always positive in your formalism, this equation is for produced metabolites only. A minus sign should be added for consumed metabolites.

Thank you for pointing this out. F is indeed always positive, and equation 5 only described produced metabolites. Equation 5 has been corrected to also include the consumed metabolites. We have made the same correction to equation 2 and added clarifications to the surrounding text.

References

- [1] L. De Vuyst et al. “Summer Meeting 2013: Growth and physiology of bifidobacteria”. In: *Journal of Applied Microbiology* 116.3 (2014), pp. 477–491. ISSN: 13645072. DOI: 10.1111/jam.12415.
- [2] Milan J.A.van Hoek and Roeland M.H. Merks. “Emergence of microbial diversity due to cross-feeding interactions in a spatial model of gut microbial metabolism”. In: *BMC Systems Biology* 11.1 (2017), pp. 1–18. ISSN: 17520509. DOI: 10.1186/s12918-017-0430-4.
- [3] François Pierre J. Martin et al. “Impact of breast-feeding and high-and low-protein formula on the metabolism and growth of infants from overweight and obese mothers”. In: *Pediatric Research* 75.4 (2014), pp. 535–543. ISSN: 15300447. DOI: 10.1038/pr.2013.250.
- [4] Van T. Pham et al. “Early colonization of functional groups of microbes in the infant gut”. In: *Environmental microbiology* 18.7 (2016), pp. 2246–2258. ISSN: 14622920. DOI: 10.1111/1462-2920.13316.
- [5] Kathleen Sim et al. “Dysbiosis anticipating necrotizing enterocolitis in very premature infants”. In: *Clinical Infectious Diseases* 60.3 (2015), pp. 389–397. ISSN: 15376591. DOI: 10.1093/cid/ciu822.
- [6] E. M. Stansbridge et al. “Effects of feeding premature infants with *Lactobacillus* GG on gut fermentation”. In: *Archives of Disease in Childhood* 69.5 SPEC NO (1993), pp. 488–492. ISSN: 00039888. DOI: 10.1136/adc.69.5.Spec_No.488.
- [7] Meyer J. Wolin et al. “Changes in Production of Ethanol, Acids and H₂ from Glucose by the Fecal Flora of a 16- to 158-d-Old Breast-Fed Infant”. In: *The Journal of Nutrition* 128.1 (1998), pp. 85–90. ISSN: 0022-3166. DOI: 10.1093/jn/128.1.85.

August 2, 2022

Prof. Roeland M.H. Merks
Institute of Biology, Leiden University
Animal Sciences & Health
Einsteinweg 55
Leiden 2333 CC
Netherlands

Re: mSystems00446-22R1 (A multiscale spatiotemporal model including a switch from aerobic to anaerobic metabolism reproduces succession in the early infant gut microbiota)

Dear Prof. Roeland M.H. Merks:

I am pleased to inform you that your manuscript has been accepted. Please implement the last request of the reviewer when checking the proofs.

For your reference, ASM Journals' address is given below. Before it can be scheduled for publication, your manuscript will be checked by the mSystems production staff to make sure that all elements meet the technical requirements for publication. They will contact you if anything needs to be revised before copyediting and production can begin. Otherwise, you will be notified when your proofs are ready to be viewed.

Publication Fees:

If you would like to submit a potential Featured Image, please email a file and a short legend to msystems@asmusa.org. Please note that we can only consider images that (i) the authors created or own and (ii) have not been previously published. By submitting, you agree that the image can be used under the same terms as the published article. File requirements: square dimensions (4" x 4"), 300 dpi resolution, RGB colorspace, TIF file format.

We recognize that the video files can become quite large, and so to avoid quality loss ASM suggests sending the video file via <https://www.wetransfer.com/>. When you have a final version of the video and the still ready to share, please send it to mSystems staff at msystems@asmusa.org.

Sincerely,

Karoline Faust
Editor, mSystems

Journals Department
Video S3: Accept
Video S2: Accept
Figure S2: Accept
Video S4: Accept
Figure S1: Accept
Figure S3: Accept
S2 Table: Accept
Figure S4: Accept
Video S1: Accept
S1 Table: Accept